# Rapid synthesis of micron-thick flexible graphite films via non-equilibrium carbon flux engineering

Haiyang Liu[1,2,8], Zhen Wang[2,8], Xu Wang[1,2,8], Jiayu Chang[2], Xinfu Hou[3], Linxuan Li[4], Mengyuan Liu[5], Xiongzhi Zeng[6], Qi Cai[1,2], Qingyu Zhou[2], Junwei Deng[2,7], Chengjin Wu[2], Sicong Zheng[2,7], Zhenyu Li[6], Mengxi Liu[5], Wu Zhou[4], Bo Sun[3]✉, Luzhao Sun[1,2]✉ & Zhongfan Liu[1,2]✉

The scalable synthesis of high-quality graphite materials remains a formidable challenge due to the inherent trade-off between crystalline perfection and manufacturing efficiency. Existing forms of graphite, such as highly oriented pyrolytic graphite (HOPG) and Kish graphite, suffer from sluggish pyrolytic processes, limited carbon diffusion rates and energy-intensive protocols, often requiring several days for production. Here, we report a pulsed Joule heating-induced carburization (PJHIC) strategy that exploits transient non-equilibrium states to enable rapid carbon diffusion and segregation in metal substrates. By applying instantaneous thermal shocks ($>1300\,°C$, $>300\,°C/s$ heating rate) to solid carbon precursor-coated nickel and cobalt foils, we demonstrate the rapid carbon transport in bulk metals and achieve a vertical graphite growth rate of 730 nm/min, which is an order of magnitude faster than conventional methods. Cyclic temperature pulses further enable the synthesis of 1–5 μm-thick ABA-stacked graphite films with millimeter-scale grain sizes. The resulting rapid epitaxially grown graphite films exhibit a highly ordered crystalline structure and exceptional thermal conductivity ($1314\,W\,m^{-1}\,K^{-1}$), comparable to high-quality HOPG and Kish graphite. This work establishes a non-equilibrium synthesis paradigm for high-quality layered materials, bridging atomic-scale precision with industrial-scale manufacturing.

Graphite, a cornerstone material in both industrial engineering and fundamental physics, has garnered enduring acclaim for its exceptional thermal conductivity ($>1500\,W\,m^{-1}\,K^{-1}$) and extraordinary high-temperature stability ($>3000\,°C$ sublimation point)[1,2]. Its layered van der Waals architecture has also enabled groundbreaking discoveries in band structure engineering and quantum Hall effect research[3–5]. Moreover, graphite thinned to micron dimensions ($<10\,μm$) gains mechanical compliance while sustaining crystallinity, positioning it as

[1]Center for Nanochemistry, Beijing Science and Engineering Center for Nanocarbons, Beijing National Laboratory for Molecular Sciences, College of Chemistry and Molecular Engineering, Peking University, Beijing, China. [2]Technology Innovation Center of Graphene Metrology and Standardization for State Market Regulation, Beijing Graphene Institute, Beijing, China. [3]Tsinghua SIGS, Tsinghua University, Shenzhen, China. [4]School of Physical Sciences, University of Chinese Academy of Sciences, Beijing, China. [5]CAS Key Laboratory of Standardization and Measurement for Nanotechnology, National Center for Nanoscience and Technology, Beijing, China. [6]Key Laboratory of Precision and Intelligent Chemistry, University of Science and Technology of China, Hefei, China. [7]College of Energy, Soochow Institute for Energy and Materials Innovations (SIEMIS), Key Laboratory of Advanced Carbon Materials and Wearable Energy Technologies of Jiangsu Province, Soochow University, Suzhou, China. [8]These authors contributed equally: Haiyang Liu, Zhen Wang, Xu Wang. ✉e-mail: sun.bo@sz.tsinghua.edu.cn; sunlz-cnc@pku.edu.cn; zfliu@pku.edu.cn

an ideal candidate for next-generation flexible thermal spreaders, electrothermal actuators, EUV manufacturing shielder, and conformal electronic skins[6–12].

However, the scalable synthesis of high-quality graphite materials remains constrained by an irreconcilable trade-off between crystalline quality and production efficiency. Conventional approaches face fundamental limitations. For instance, highly oriented pyrolytic graphite (HOPG) synthesis suffers from slow pyrolysis process ($\sim$0.1 μm/h deposition rate, >24 h cycle time), sustained high temperature and pressure (2500–3000 °C, 30 MPa), and limited size (<10 cm²)[1,2,13]. The Kish graphite generates micron-sized grains (1–5 μm) through rapid melt solidification, necessitating 2000 °C post-annealing to reduce porosity (>30%) and metal impurities (500–1000 ppm Fe)[2,14]. Polymer-derived graphitization struggles with energy-intensive processing (3000 °C for days), poor crystallinity, and difficulty in thickness control below 10 μm[2,15].

Transition metal (e.g., Ni, Co) catalyzed epitaxy offers a promising alternative by enabling continuous graphene layer segregation at reduced graphitization temperatures (1000$\sim$1300 °C)[16–21]. These metal (foil) substrates also act as templates to form graphite films. However, current approaches remain limited by the amount of dissolved carbon and insufficient carbon diffusion during continuous epitaxy, resulting in either submicron thickness or prohibitively long processing times (over days)[20–30].

The diffusion of carbon species in metals is governed by two driving forces: (1) concentration gradient-driven diffusion under isothermal conditions, and (2) forced segregation caused by cooling-induced reductions in carbon solubility. The latter provides significantly stronger driving forces and faster carbon transport. Capitalizing on this, we propose a non-equilibrium diffusion strategy leveraging rapid thermal cycling. Using pulsed Joule heating technique[31–33] to achieve rapid temperature ramping and cooling, we revealed the accelerated carbon diffusion during the forced segregation process (>4.5 μm/s), enabling the rapid synthesis of high-quality, micron-thick graphite films with a vertical growth rate of 730 nm/min on Ni, which is an order of magnitude faster than conventional methods. Cyclic temperature pulses further overcome solubility limits, producing 1–5 μm-thick ABA-stacked graphite films with millimeter-sized grains. The resulting films exhibit a highly ordered crystalline structure, with interlayer spacings (3.355 Å) and thermal conductivity (1314 W m⁻¹ K⁻¹) comparable to those of high-quality HOPG and Kish graphite. This work establishes non-equilibrium carbon-flux engineering as a general paradigm for layered-material synthesis.

## Results and discussion
### Pulsed Joule heating activated rapid carbon diffusion and segregation

Figure 1a illustrates the pulsed Joule heating-induced carburization (PJHIC) strategy for rapid synthesis of graphite films. Polymethyl methacrylate (PMMA) serves as solid-state carbon source, coated on a nickel foil (50 μm-thick) and in a sealed reaction chamber. Application of pulsed current through graphite electrodes generates instantaneous Joule heating, elevating the system to 1300 °C with heating rate exceeding 300 °C/s (Fig. 1b and Supplementary Fig. 1). This heating enables thermal decomposition of PMMA, releasing carbon species that diffuse into the nickel matrix. Subsequent annealing promotes carbon diffusion under concentration gradients, while controlled cooling ($\sim$70 °C/s via water-cooling) triggers forced segregation through reduced solubility of nickel, forming high-quality graphite films (Fig. 1c). Figure 1d, e show as-synthesized graphite films transferred onto Si/SiO₂ and flexible polyethylene terephthalate (PET) substrates, demonstrating micron-scale thickness and flexibility with continuous, crack-free coverage on curved PET.

To investigate the growth rate limit of graphite segregation through Ni foil, PJHIC experiments with varying annealing time were performed (Fig. 1f). A three-stage pulsed current protocol (240 A → 320 A → 280 A) mitigated thermal overshoot, maintaining ideal thermal plateau with consistent ramp/cool rates. Given that 1000 °C is the conventional growth temperature for graphene[17,24], the effective growth durations in our experiments are defined as the period above 1000 °C, which are calculated to be 11 s, 42 s, 51 s, 61 s, 72 s, 102 s, and 132 s, respectively. Figure 1g (Details please see in Supplementary Figs. 2 and 3) shows the resulting film thickness evolution, in which graphite nucleated within 11 s, thickened progressively, and saturated at $\sim$800 nm beyond 72 s. The rapid nucleation indicates non-equilibrium carbon diffusion through nickel bulk, consisting with Fickian diffusion models (Supplementary Note 1)[34]. With a growth duration of 61 s, a maximum vertical growth rate was calculated to be 730 nm/min (Supplementary Note 2), outperforming state-of-art methods that rely on prolonged thermal treatments (<100 nm/min) (Fig. 1h). Notably, considering that graphite segregation occurs predominantly during the rapid cooling stage (from 1300 °C to 1000 °C within 4 - 5 seconds), the corresponding instantaneous segregation rate is calculated to be $\sim$12,000 nm/min (Supplementary Fig. 4). The PJHIC strategy enables micron-thick, uniform graphite films with growth rates over an order of magnitude higher than previous reports[15,20–29,35–42].

Raman spectroscopy of as-synthesized graphite films with different growth time (Fig. 1i and j) revealed prominent G ($\sim$1580 cm⁻¹) and 2D ($\sim$2670 cm⁻¹) bands with negligible D-band intensity, indicating low defect density. Deconvolution of the 2D band into asymmetric Lorentzian components centered at 2680 cm⁻¹ ($2D_1$) and 2718 cm⁻¹ ($2D_2$) confirms AB stacking order of the graphene layers[43,44]. Spatial mapping across an 80×80 μm² area, coupled with statistical analysis of >10,000 spectra (Fig. 1k and Supplementary Fig. 5) yields an average $I_D/I_G$ ratio of $\sim$0.03 and G-band full width at half maximum (FWHM) of 13 cm⁻¹, quantitatively validate the exceptional crystallinity of the as-synthesized graphite films[45].

To gain deeper insights into the spatial evolution of carbon distribution, time-of-flight secondary ion mass spectrometry (ToF-SIMS) characterizations along with depth-resolved analysis[46,47] were conducted on pristine Ni foil and PJHIC-samples with different growth durations of 10, 12, and 72 s. The 3D distributions of carbon and Ni elements were reconstructed by detecting the secondary ions of C⁻ and Ni⁻, respectively. The pristine Ni sample shows negligible carbon signals (Fig. 2a, b), while the 10-second-grown sample exhibits homogeneous C⁻ signal enhancement throughout the Ni matrix (Fig. 2c, d), confirming bulk carbon dissolution prior to segregation onset. Crucially, a graphite layer emerged at 12-second-grown sample (Fig. 2e, f), directly evidencing that carbon segregation initiates within the critical 10-12 s window and revealing rapid phase transformation kinetics. Prolonged growth to 72 s yields continuous graphite films averaging $\sim$800 nm thickness (Fig. 2g and Supplementary Fig. 3d). Strikingly, the reconstructed 3D map revealed localized thickening specifically within Ni grain boundary grooves (Fig. 2h), contrasting with the 800 nm baseline in adjacent regions. This boundary-confined enhancement demonstrates accelerated carbon migration along grain defects, where the reduced atomic packing density lowers the diffusion barriers, the defects enable preferential nucleation and growth.

Notably, at graphite/Ni interfaces, the pseudomorphic nickel enrichment signature results from increased secondary Ni⁻ yields due to altered electronic states at imperfectly bonded interfaces (Fig. 2e–h and Supplementary Fig. 6). Cross-sectional transmission electron microscopy (TEM) combined with energy-dispersive X-ray spectroscopy (EDS) analysis independently validated the clear graphene/Ni interface structure on the 72-second growth sample, where C signals terminate coincidently with Ni signal onset within < 5 nm interfacial width (Supplementary Fig. 7). The multiscale analysis conclusively establishes that non-equilibrium thermal shocks drive explosive segregation rates by synergizing bulk carbon super-saturation with boundary-mediated rapid nucleation.

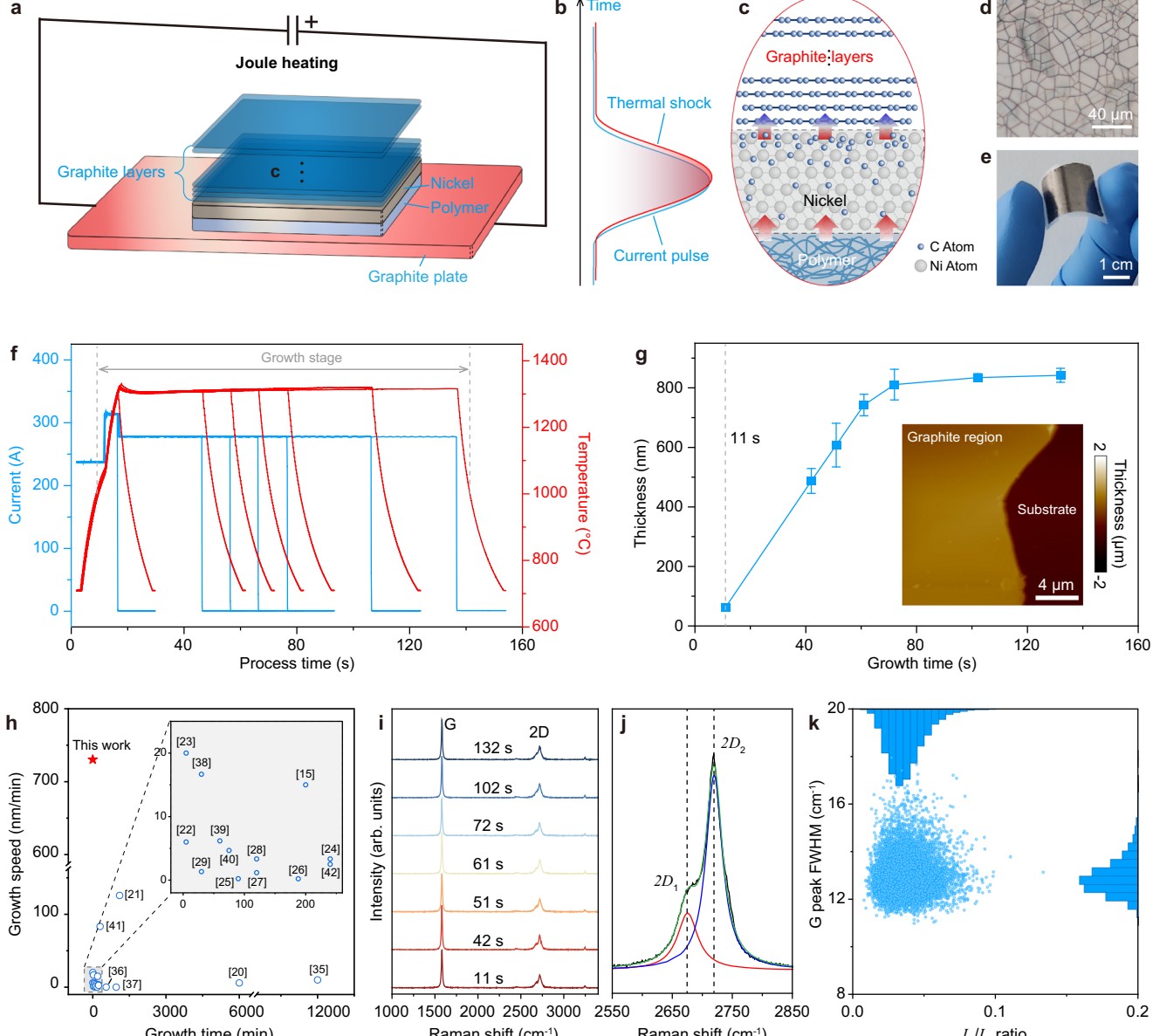

**Fig. 1 | Rapid growth of graphite films via the PJHIC process. a–c** Schematic of the PJHIC system (**a**), Schematic diagram of the thermal shock generated by pulsed current application (**b**), and carbon diffusion–graphite segregation process (**c**). This process primarily relied on carbon supplied by the thermal decomposition of PMMA; after diffusion of carbon within the Ni foil, rapid cooling triggered carbon segregation and precipitation, ultimately yielding high quality graphite films. **d, e** Optical images of the as-synthesized graphite film transferred on Si/SiO₂ (**d**) and flexible PET (**e**) substrate, respectively. **f** Current–time and temperature–time profiles corresponding to different growth durations. **g** Graphite thickness as a function of growth time. Inset: AFM image at side of a typical graphite sheet grown with 72 s. The error bars represent the standard deviation of repeated experiments. **h** Comparative analysis of the growth time and growth rate of graphite films reported in this work and in literatures[15,20–29,35–42]. Inset: magnified view of the lower-left region. **i** Raman spectra acquired at as-synthesized graphite films with different growth time. **j** Deconvolution of a typical 2D Raman band into two Lorentzian components. **k** The statistical distribution of $I_D/I_G$ ratio and FWHM of the G band over an 80 × 80 μm² region.

The rapid non-equilibrium segregation growth process comprises two distinct yet interconnected stages (Supplementary Note 3). In Stage I (Solute depletion-driven carbon precipitation), the rapid cooling from 1300 °C to 1000 °C within 4 ~ 5 seconds creates a massive carbon supersaturation (Δ$C$ ≈ 0.28 wt.%) due to the drastic reduction in equilibrium solubility dictated by the Ni-C phase diagram[34]. This corresponds to a chemical potential difference of Δ$\mu$ ≈ 0.08 eV/atom. In Stage II, the accumulated carbon overcomes a drastically reduced nucleation barrier and forms graphite layers. Classical nucleation theory indicates that the high Δ$\mu$ lowers both the critical nucleation radius $r^*$ and energy barrier Δ$G^*$, facilitating the observed high-density nucleation and rapid graphite layer formation[48,49]. Quantitatively, the

carbon atomic flux in PJHIC reaches $F_{PJHIC}$ ≈ 2.1 × 10²² atoms·m⁻²·s⁻¹, surpassing state-of-the-art CVD graphene growth by two orders of magnitude ($F_{CVD}$ ≈ 3.1 × 10²⁰ atoms·m⁻²·s⁻¹)[50,51]. This fundamental difference in mass transport efficiency underscores the paradigm-shifting nature of the non-equilibrium forced segregation mechanism compared to diffusion-limited processes.

## Cyclic saturation engineering for thickness control
To transcend the sub-micrometer thickness constraint of single-step graphite growth, we engineered a cyclic heating–cooling strategy harnessing carbon segregation kinetics (Fig. 3a) with upgraded carbon source and extended metal substrates. Theoretical modeling

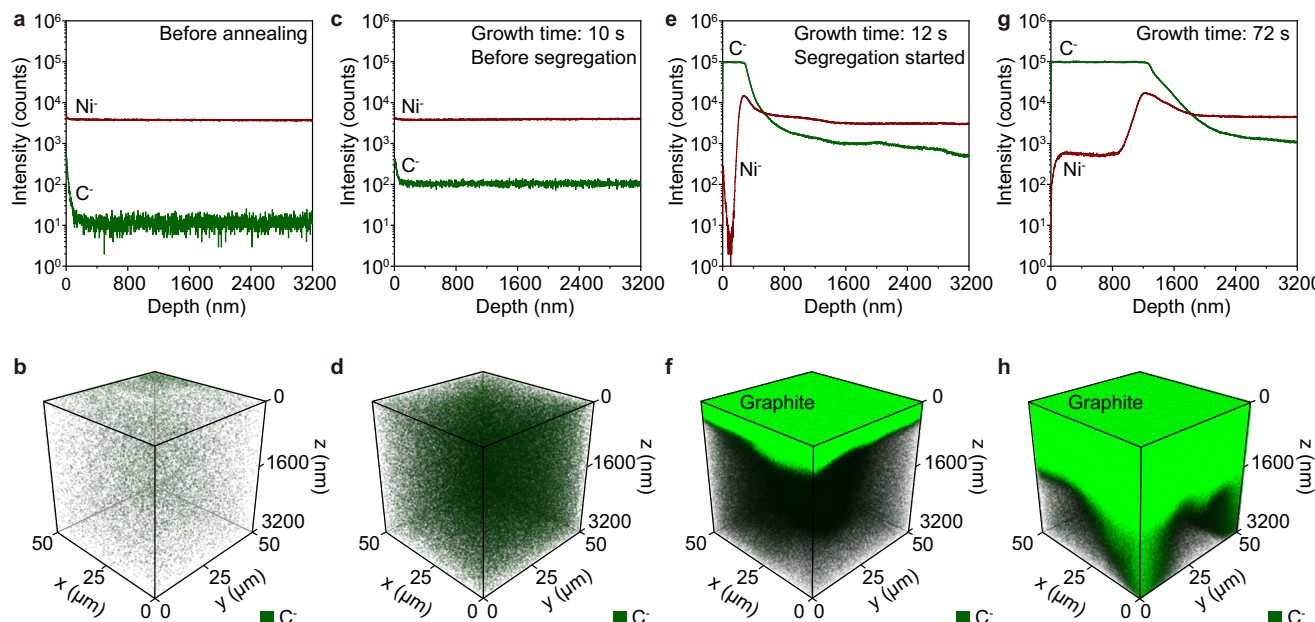

**Fig. 2 | ToF-SIMS characterizations of graphite samples with different growth times. a, b** Depth profiles of Ni⁻ and C⁻ secondary ion signals (**a**) and corresponding 3D distribution map of C⁻ intensity (**b**) obtained from pristine Ni foil. **c, d** Depth profiles (**c**) and 3D map from 10 s sample. **e, f** Depth profiles (**e**) and 3D map (**f**) from 12 s sample. **g, h**, Depth profiles (**g**) 3D map (**h**) from 72 s sample.

established a maximum segregation thickness of 1.17 μm (Supplementary Note 1), while experimental optimization of PMMA volume (0.2–0.8 mL) yielded 1.07 ± 0.03 μm at 0.6 mL (Supplementary Fig. 8). Although the 7.42% carbon loss happens due to PMMA decomposition into gaseous byproducts (Supplementary Note 4 and Supplementary Table 1), the decomposed gaseous carbon source favors a rapid and massive carbon influx into the Ni substrate, potentially creating a higher initial supersaturation. The cyclic process (Fig. 3a) enforced continuous chemical potential gradients through precisely controlled thermal pulses: each cycle consisted of 120 s of high-current heating for dissolving carbon, followed by 30 s of forced cooling for triggering the segregation (Supplementary Fig. 9).

This approach achieved programmable thickness control (1-5 μm) on both Ni and Co foils through 1-50 cycles (Fig. 3b). Figure 3c shows the step edges of the graphite layers grown on the Ni surface, characterized by white light interferometry (WLI), which allow the measurement of the graphite film thickness (Fig. 3d). Furthermore, we statistically analyzed the thickness of graphite layers segregated from the surfaces of Ni and Co foil (Supplementary Fig. 10), revealing a linear growth relationship between thickness and the number of cycles (Fig. 3e, f). Notably, the vertical growth rate of ~0.1 μm per cycle (150 s) was lower than the single-step process due to partial redissolution of low-crystallinity layers during reheating. Nevertheless, a 5 μm-thick film requires only ~2 h total processing, demonstrating a several times throughput increase over conventional CVD, and demonstrating the strong capability for the rapid synthesis of microscale-thick graphite films.

### Characterization of as-obtained graphite films
The crystallographic orientation and domain size of the as-obtained graphite films were investigated by electron backscatter diffraction (EBSD) characterizations (Fig. 4a, b and Supplementary Fig. 11). The inverse pole figure (IPF) map in the out-of-plane (Z) exhibits a globally uniform red intensity, confirming consistent (0001) stacking orientation across the film. The in-plane (Y) IPF mapping reveals > 1.2 mm domain size with > 95% blue areas signifying dominant single in-plane orientation, while cyan-to-green grains (< 5%) indicate minor misoriented grains with grain boundaries. This domain size substantially

exceeds that of the underlying Ni substrate (100–200 μm, Supplementary Fig. 11a), suggesting graphene's rapid epitaxial propagation across substrate grain boundaries during segregation. The in-plane alignment of the graphite films was predominantly governed by intrinsic graphene layer orientation rather than the Ni lattice. The black vein-like regions observed in the inverse pole figures correspond to wrinkle structures, where three-dimensional distortion led to the loss of Kikuchi diffraction signals. It is noteworthy that although films synthesized on single-crystalline Ni(111) foils exhibited a single crystallographic orientation, their thickness was reduced due to the absence of grain boundary acceleration effects; the unresolved EBSD area is even more due to the stronger graphite-Ni(111) interaction and thus resulting wrinkles (Supplementary Fig. 11e–h). Owing to the large domain size and high continuity, the electrical conductivity reaches 6.73 × 10⁵ S·m⁻¹ with high uniformity (Supplementary Fig. 12).

The graphitization degree was evaluated by using X-ray diffraction (XRD) (Fig. 4c). The as-obtained PJHIC-graphite film exhibits (0002) peak at 26.543°, with a full width at half maximum (FWHM) of 0.169°. This peak position was compared with those of several commercially available graphite: HOPG (26.511°, FWHM = 0.173°), natural graphite (26.562°, FWHM = 0.167°), and Kish graphite (26.553°, FWHM = 0.165°). According to the Bragg's law, the interlayer spacing of the PJHIC-graphite film is calculated as 3.355 Å, closely matching that of HOPG (3.360 Å), natural graphite (3.354 Å) and Kish graphite (3.355 Å).

Atomic-scale characterization via scanning tunneling microscopy (STM) reveals continuous hexagonal lattices (Fig. 4d). TEM combined with selected area electron diffraction (SAED) characterizations show hexagonal symmetric patterns, and the intensity ratios $I_{1-210}/I_{0-10}$ and $I_{2110}/I_{1010}$ is about 4, indicative of ABA-stacking structure (Fig. 4e)[52]. Angular dark-field (ADF) scanning transmission electron microscopy (STEM) image and corresponding Fast Fourier transform (FFT) patterns also match the simulated ABA-stacked registry (Fig. 4f and Supplementary Fig. 13). Cross-sectional HRTEM confirmed that the basal-plane C-C bonds of 0.246 nm and c-axis periodicity of 0.68 nm (Fig. 4g, h).

In-plane ($\Lambda_{//}$) and cross-plane ($\Lambda_{\perp}$) thermal conductivity were measured via time-domain thermoreflectance (TDTR) technique[53–55], yielding 1314 W m⁻¹ K⁻¹ and 7.51 W m⁻¹ K⁻¹, respectively (Fig. 4i, j). These

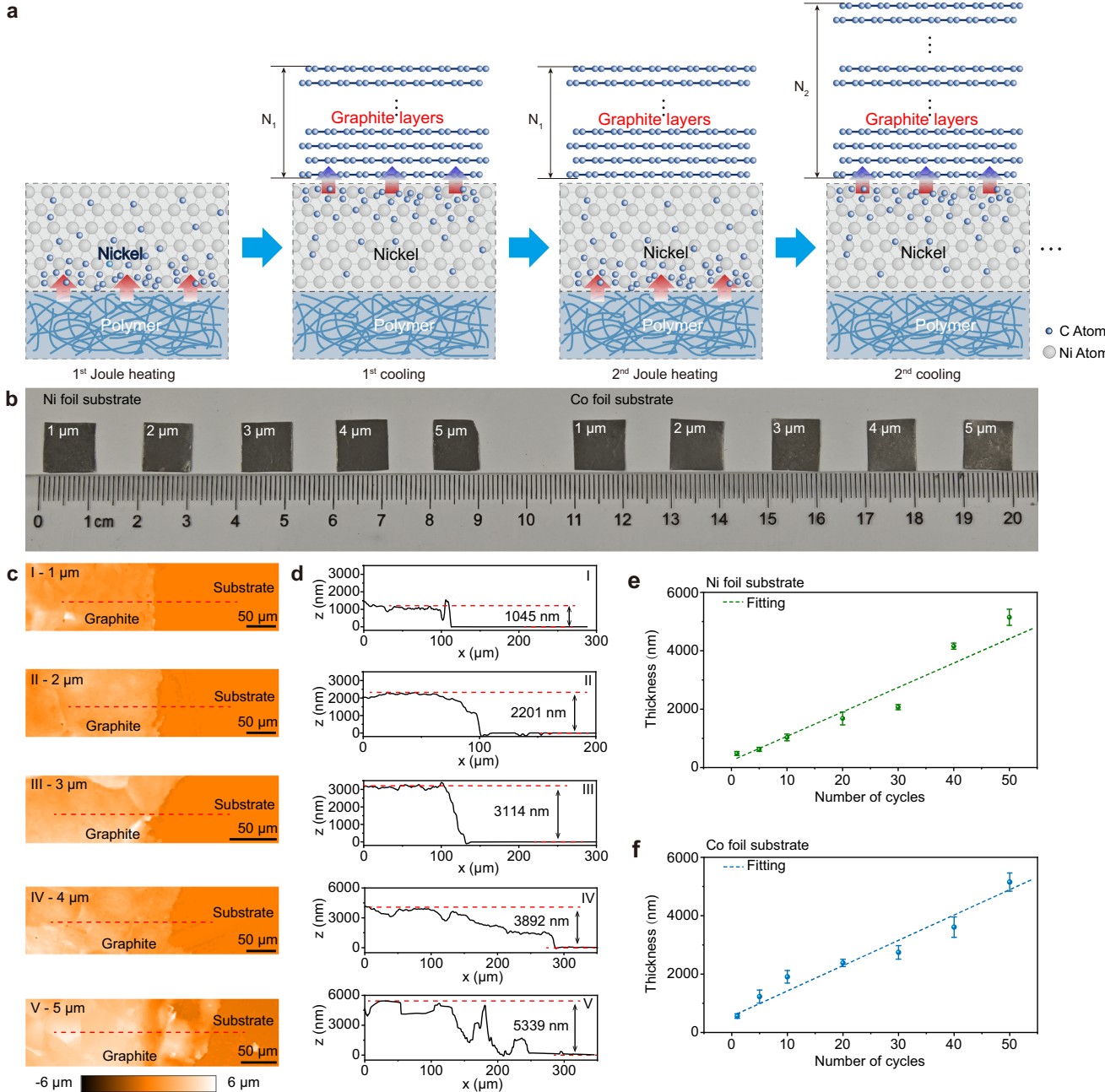

**Fig. 3 | Micron-thick graphite film thickness control via cyclic saturation engineering. a** Schematic illustration of cyclic heating–cooling–induced repeated carbon segregation. **b** Optical images of graphite films with various thicknesses on Ni and Co foils. **c** WLI maps at the graphite film edges with thicknesses ranging from 1 to 5 μm. **d** Cross-sectional height profiles extracted from WLI maps of (**c**). **e, f** Scatter plot and linear fitting of graphite thickness as a function of cycle number on Ni foil (**e**) and Co (**f**) foil, respectively. The error bars represent the standard deviation of repeated experiments.

values are comparable to high-quality HOPG ($\Lambda_{//} = 1480$ W m$^{-1}$ K$^{-1}$, $\Lambda_{\perp} = 8.5$ W m$^{-1}$ K$^{-1}$) and Kish graphite ($\Lambda_{//} = 1362$ W m$^{-1}$, $\Lambda_{\perp} = 8.57$ W m$^{-1}$ K$^{-1}$), and substantially exceed natural graphite ($\Lambda_{//} = 904$ W m$^{-1}$ K$^{-1}$, $\Lambda_{\perp} = 8$ W m$^{-1}$ K$^{-1}$), (Fig. 4k and Supplementary Figs. 14−15). These results demonstrate that fast-epitaxially grown-graphite films by the PJHIC process exhibit structural characteristics matching those of ideal graphite and achieve exceptional crystallinity.

In summary, this work demonstrates a pulsed Joule-heating-induced carburization method that enables rapid synthesis of high-quality, micron-thick graphite films by leveraging transient non-equilibrium thermal states. Scientifically, we reveal an exceptionally high rate of carbon transport (> 4.5 μm/s) within bulk metals during cooling-driven segregation, for which the driving force is

significantly greater than that of steady-state diffusion. We achieved a record vertical growth rate of 730 nm/min for ABA-stacked graphite films, which exhibit millimeter-scale domain size and excellent thermal conductivity of 1314 W m$^{-1}$ K$^{-1}$, rivaling commercial HOPG and Kish graphite. This approach marks a substantial advance in synthesis speed, but the crystalline perfection of the resulting films still falls short of single-crystal standards, as the rapid non-equilibrium process introduces grain boundaries, point defects, and wrinkles. Our insights into segregation dynamics suggest that further optimization of kinetic control could reconcile both speed and quality. The PJHIC method also shows good scalability, as demonstrated by a large-area graphite film with a size of 12 cm × 5 cm (Supplementary Fig. 16). We anticipate that non-equilibrium carbon

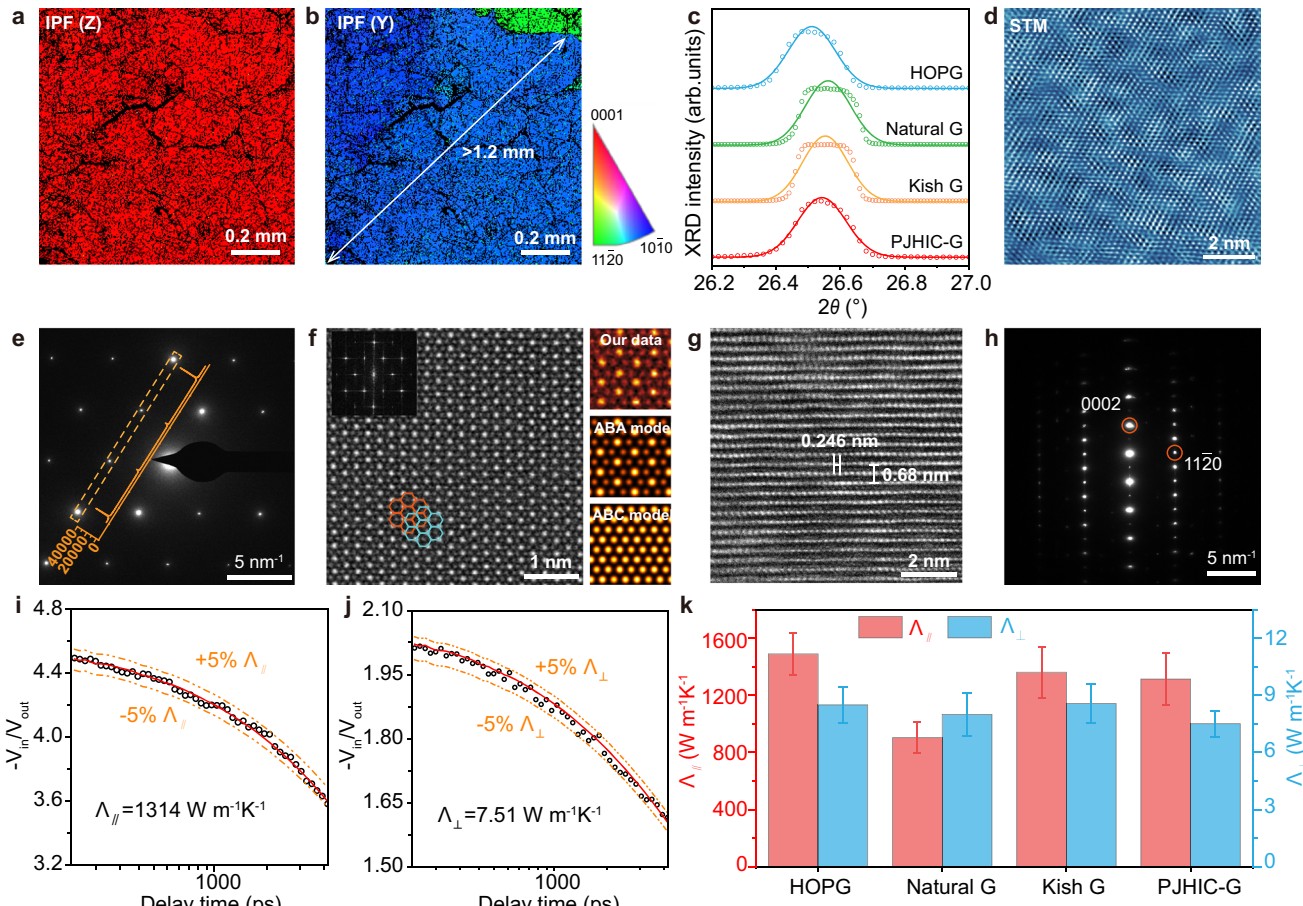

**Fig. 4 | Characterizations of as-obtained graphite films. a, b** EBSD IPF maps along the Z (**a**) and Y (**b**) axes of PJHIC-graphite films on a polycrystalline Ni foil. The Inverse Pole Figure legend applies to both panels (**a**) and (**b**). **c** High-resolution XRD 2θ scans of the (0002) graphite peak collected from HOPG, natural graphite, Kish graphite and the as-obtained PJHIC-graphite, respectively. **d** STM image. **e** SAED patterns. Inset: Intensity profile along the orange line. **f** ADF-STEM image and corresponding simulated ABA and ABC models; inset: corresponding FFT pattern. The orange and cyan hexagonal markers denote different layers of graphite. **g, h** Cross-sectional HRTEM image (**g**) and corresponding electron diffraction patterns (**h**). **i, j** TDTR data and fitting analysis of the in-plane (**i**) and cross-plane (**j**) thermal conductivity of PJHIC-graphite film, respectively. Thermal conductivity of graphite extracted by fitting the TDTR data ($V_{in}/V_{out}$) using a thermal diffusion model (parameters are listed in Supplementary Table 2). **k** Thermal conductivity comparison of four types of graphite. The error bars represent the standard deviation of repeated experiments.

flux engineering will provide a generalizable pathway for the high-throughput synthesis of other layered materials requiring precise atomic control and industrial-scale viability.

## Methods

### PJHIC growth
High-purity nickel foil (purity 99.9%, thickness 50 μm) and cobalt foil (purity 99.9%, thickness 50 μm) were commercially obtained from Beijing Gaoke New Materials Technology Co., Ltd. Poly(methyl methacrylate) (PMMA, Mw ≈ 950 kDa) was supplied by KAYAKU Advanced Materials, Inc. (model: PMMA 950 K A4, spin-coating speed 4000 rpm).

The Ni/Co foils were cut into 21 mm × 21 mm pieces. Subsequently, approximately 0.6 mL of PMMA solution was spin-coated onto the surface of the foils. Based on a systematic evaluation of decomposition kinetics and comparative experiments with alternative sources (e.g., graphite paper, carbon black powder, and PI film), PMMA is chosen as the carbon source suitable for the rapid PJHIC process. In the cyclic growth experiments, the PMMA carbon source was coated only once at the initial stage and the PMMA-coated sample was directly placed on graphite paper for the subsequent cyclic graphitization process. The coated samples were then cured at 170 °C for 10 min and

placed inside a Joule heating chamber (JH3.3-P type, Hefei In Situ Technology Co., Ltd.). Prior to heating, the chamber was purged three times with argon gas, each cycle consisting of pressurization to 0.3 MPa followed by evacuation. Synthesis was conducted under a constant argon flow with a pressure of approximately 0.1 MPa and a flow rate of 2000 sccm. The samples were rapidly heated to 1300 °C under a three-stage current program: 240 A for initial rapid heating, 320 A to reach the peak temperature of 1300 °C, and 280 A for isothermal holding at 1300 °C throughout the growth period. After growth. For the cyclic process, a current of 320–325 A was applied to the graphite plate for 120 s to rapidly achieve 1250–1280 °C, ensuring sufficient dissolution and minimizing heat accumulation during repeated cycling. The cooling stage was set to be 30 s to promote effective carbon segregation.

### Transfer of graphite films
The experimentally obtained graphite/nickel films were cut into samples measuring 20 mm × 20 mm. Subsequently, the nickel substrate was removed by wet chemical etching: the samples were immersed in an aqueous ferric chloride ($FeCl_3$) solution (1 mol/L concentration) for more than 10 h to obtain self-supporting graphite films. The etched films were repeatedly rinsed with deionized

water (3-4 times) to remove residual metal ions adsorbed on the surface. Finally, the graphene films were transferred onto silicon dioxide/poly(ethylene terephthalate) ($SiO_2$/PET) substrates and dried at room temperature to eliminate residual moisture between the film and the substrate, preparing them for subsequent characterization.

## ToF-SIMS measurements

Time-of-flight secondary ion mass spectrometry (ToF-SIMS) analysis of freshly isolated trilayer graphene island samples was performed using an ION ToF-SIMS 5 instrument (Germany). Surface sputter depth profiling was initially conducted with a 2 keV $Cs^{1+}$ primary ion beam, followed by compositional imaging using a 30 keV $Bi^{1+}$ primary ion beam. Imaging areas of $50 \times 50$ $\mu m^2$ were acquired with an image resolution set to $128 \times 128$ pixels. Under these conditions, the spatial resolution in the XY plane−defined as the distance over which the signal intensity at the graphene domain boundary rises from 25% to 75%−was approximately 0.2 μm, while the depth resolution was approximately 1 nm. The entire analysis was carried out under a chamber-based vacuum of approximately $10^{-8}$ mbar. Final data processing, including characteristic peak identification and image reconstruction, was performed using the instrument's proprietary SurfaceLab 7.0 software (ION-TOF).

## STEM Characterizations

STEM characterization was performed using an aberration-corrected JEOL GRANDARM2 microscope operated at 200 kV. ADF-STEM image was acquired with a beam convergence semi-angle of 32 mrad and a collection semi-angle range of 36−147 mrad. The image was subsequently denoised using an average background-subtracted filter.

## Thermal conductivity measured by TDTR

TDTR is a non-contact optical method for characterizing the thermal transport properties of various materials. In the methodology, an ultrafast femtosecond laser source (785 nm, 80 MHz repetition rate, linearly polarized) was split into pump and probe beams using a polarizing beam splitter. The pump beam was modulated at radio frequencies via an Electro-Optical Modulator to induce periodic thermal oscillations at the sample surface. Concurrently, the probe beam− modulated at 200 Hz with a mechanical chopper−detected temperature-dependent reflectance variations. The 600 mm delay stage can control the time difference between the pump beam and the probe beam reaching the sample. Prior to measurements, an Aluminum layer (109 nm thickness, deposited by magnetron sputtering; thickness determined by picoacoustics) was applied to the sample surface as transducer, when the pump beam is directed onto the Aluminum, a fraction of its energy is absorbed by the film, causing the surface temperature to rise. Consequently, the reflected probe-beam intensity varies directly with the temperature change. The photodetector converts an optical signal into an electrical signal. Acquired signals were demodulated relative to the pump-probe delay time using lock-in amplifiers.

By employing objective lens configurations with selective sensitivity to the direction of heat transport (a 5× objective provide greater sensitivity to the cross-plane thermal conductivity, $\Lambda_\perp$, whereas a 20× objective is more sensitive to the in-plane thermal conductivity, $\Lambda_{//}$), and conducting measurements at the same location and under an identical pump modulation frequency of 10.1 MHz, the independent thermal conductivity components were successfully separated and obtained. In this work, the measured HOPG is from Tipsnano Co., Ltd. (ZYB grade). Although premium HOPG (e.g., SPI Co., Ltd., with $0.8° \pm 0.2°$ mosaic spread) achieves higher $\Lambda_{//}$ (2010 W m$^{-1}$ K$^{-1}$) and $\Lambda_\perp$ (13.4 W m$^{-1}$ K$^{-1}$), the measured Tipsnano HOPG properties align with its specified structural parameters[55].

## Data availability

Relevant data supporting the key findings of this study are available within the article and its Supplementary Information file. Source data are provided with this paper. Further information is also available from the corresponding authors upon request. Source data are provided with this paper.

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

## Acknowledgements

This work was supported by the National Natural Science Foundation of China (No. T2188101 to Z.F. Liu and L.S.), Beijing Natural Science Foundation (No. 2264127 to L.S.), Shenzhen Science and Technology Program (Nos. RCYX20200714114643187 and WDZC20200821100123001 to B.S.), Guangdong Special Support Program (No. 2023TQ07A273 to B.S.), the CAS Project for Young Scientists in Basic Research (No. YSBR-003 to W.Z.), the Electron Microscopy Center at the University of Chinese Academy of Sciences (to W.Z.), and the Youth Innovation Promotion Association of CAS (No. 2022038 to M.X. Liu) and the CAS Project for Young Scientists in Basic Research (No. YSBR-054 to M.X. Liu).

## Author contributions

Z.F.Liu and L.S. supervised and convinced the project. L.S., H.L., and Z.W. designed the experiments and constructed the fast segregation model. H.L., Z.W., X.W., and J.C. carried out the synthesis and characterization of samples. H.L., Z.W., X.W., and L.S. analyzed the data and plotted the figures. X.H. and B.S. performed the TDTR thermal-conductivity experiments and the data analysis. L.L. and W.Z. conducted the ADF-STEM characterization and data analysis. Mengyuan Liu and Mengxi Liu conducted STM characterization and data analysis. X.Z. and Z.Y. Li helped with the two-stage model for the non-equilibrium segregation process. Q.C., Q.Z., J.D., C.W., and S.Z. helped with the figure plot, carbon diffusion model analysis and data statistics. X.W. and L.S. wrote the paper. All authors discussed the results and commented on the manuscript.

## Competing interests

The authors declare no competing interests.
