## [Peer Review file · Nature Communications]

Rapid synthesis of micron-thick flexible graphite films via non-equilibrium carbon flux engineering

Corresponding Author: Professor Zhongfan Liu

Version 0:

Reviewer comments:

Reviewer #1

(Remarks to the Author)

This manuscript presents a highly compelling and timely study on the ultrafast synthesis of high-quality, micron-thick graphite films via a pulsed Joule heating-induced carburization strategy. This work addresses a critical challenge in graphite synthesis (the trade-off between crystalline quality and manufacturing efficiency) by leveraging non-equilibrium carbon flux engineering. The authors demonstrate that transient thermal shocks enable exceptionally rapid carbon diffusion and segregation in Ni substrates, achieving a record vertical growth rate of 730 nm/min. The insights into ultrafast carbon transport offer valuable inspiration for cost-effective industrial-scale manufacturing. The work is scientifically rigorous, methodologically innovative, and well-presented, with clear potential for industrial applications. I believe it meets the high standards of Nature Communications and recommend its publication after revisions.

1. The authors demonstrated the synthesis of graphite films on centimeter-size Ni and Co foils. To further underscore the scalability and general applicability of the PJHIC method, it would be beneficial to include data on larger substrate sizes, which would help readers better assess the potential for industrial-scale production.
2. In the cyclic heating-cooling process for thickness control, it is unclear whether the same PMMA is reused across multiple cycles or recoated after each cycle. Clarifying this point in the Methods or Results section would help readers understand the carbon source consumption and the reproducibility of the cyclic growth process.
3. Although the supplementary information provides a diffusion model based on Fick's laws, a more detailed analysis incorporating the Ni–C phase diagram would strengthen the theoretical foundation. Quantitative comparisons between predicted and experimental segregation thicknesses, along with a discussion of any discrepancies (e.g., due to kinetic limitations or carbon loss), would provide deeper insight into the non-equilibrium segregation mechanism.
4. The in-plane thermal conductivity of the PJHIC-graphite film (1314 Wm⁻¹K⁻¹) is notable, but not the highest among reported graphite materials. The authors are encouraged to include a more detailed analysis linking the defect density (e.g., via Raman ID/IG ratio) and grain boundary structure (from EBSD) to the thermal conductivity. A discussion on how further defect reduction or grain orientation optimization could enhance thermal performance would be valuable for readers interested in high-thermal-conductivity applications.

Reviewer #2

(Remarks to the Author)

Version 1:

Reviewer comments:

Reviewer #2

(Remarks to the Author)

The author demonstrates a solid theoretical foundation and has effectively addressed the relevant issues. Comprehensive explanations have been provided in the revised manuscript and supplementary materials in response to the points raised earlier. Additionally, the author has supplemented detailed clarification on the migration kinetics of carbon sources within the 1000–1300°C range, elaborated on the rationale for the selection of carbon source materials and the temperature control procedures during heating and cooling, and corrected previously unclear descriptions in the characterization and analysis. This will be a commendable piece of work. I recommend that the manuscript be accepted for publication in Nature Communications.

Point-by-point response to editor and reviewers' comments

NCOMMS-25-71011-T

Editor

We sincerely appreciate the editor and reviewers' careful reading and constructive suggestions. We have thoroughly revised the manuscript according to the comments, and detailed corrections are as follows:

Response to the Reviewer #1

This manuscript presents a highly compelling and timely study on the ultrafast synthesis of high-quality, micron-thick graphite films via a pulsed Joule heating-induced carburization strategy. This work addresses a critical challenge in graphite synthesis (the trade-off between crystalline quality and manufacturing efficiency) by leveraging non-equilibrium carbon flux engineering. The authors demonstrate that transient thermal shocks enable exceptionally rapid carbon diffusion and segregation in Ni substrates, achieving a record vertical growth rate of 730 nm/min. The insights into ultrafast carbon transport offer valuable inspiration for cost-effective industrial-scale manufacturing. The work is scientifically rigorous, methodologically innovative, and well-presented, with clear potential for industrial applications. I believe it meets the high standards of Nature Communications and recommend its publication after revisions.

Authors' response:

We greatly appreciate the constructive comments provided by the reviewer 1.

Comment 1:

The authors demonstrated the synthesis of graphite films on centimeter-size Ni and Co foils. To further underscore the scalability and general applicability of the PJHIC method, it would be beneficial to include data on larger substrate sizes, which would help readers better assess the potential for industrial-scale production.

Authors' response:

We appreciate your attention to the scalability of the method and agree with your suggestion. As a preliminary exploration, we have added the demonstration of a larger graphite film with the size of 12 cm \times 5 cm (Fig. R1-1).

Fig. R1-1(Revised Supplementary Fig. 16) | Synthesis of 12 cm \times 5 cm graphite film grown on a Ni foil. (a) Optical image. (b) Typical optical microscopy image of the film. (c) Raman spectra of the graphite film.

The results indicate that continuous graphite films without macroscopic cracks were successfully fabricated on the substrate with a size of 5 cm \times 12 cm. Optical microscopy observations, together with Raman spectroscopy measurements at multiple representative points across the large-area film, confirmed good uniformity. Naturally,

achieving even larger-scale films with high quality and uniformity will require further refinement and optimization of the synthesis conditions, which will be the focus of our future work. Nevertheless, the successful synthesis on the 12 cm × 5 cm Ni foil demonstrates the significant scalability and potential for upscaling of the PJHIC method.

Accordingly, we add this figure to Supplementary Fig. 16. The corresponding revised content in the main text (in Conclusion part) is as follow:

“In summary, ... The PJHIC method also shows good scalability, as demonstrated by a large-area graphite film with a size of 12 cm × 5 cm (Supplementary Fig. 16). We anticipate that non-equilibrium carbon flux engineering will provide a generalizable pathway for the high-throughput synthesis of other layered materials requiring precise atomic control and industrial-scale viability”

Comment 2:

In the cyclic heating-cooling process for thickness control, it is unclear whether the same PMMA is reused across multiple cycles or recoated after each cycle. Clarifying this point in the Methods or Results section would help readers understand the carbon source consumption and the reproducibility of the cyclic growth process.

Authors' response:

We appreciate the reviewer's comments regarding the lack of clarity in our previous description. In the cyclic growth experiments, PMMA was applied as the carbon source only once at the initial stage and was subsequently reused in all following cycles without any re-coating. Specifically, the PMMA-coated sample was directly placed onto a graphite paper support for the subsequent treatments. During multiple rounds of carburization and carbon precipitation, the PMMA was progressively consumed; therefore, the effective carbon supply originated not only from the PMMA but also from the underlying graphite paper.

We have added this crucial experimental detail to the Methods section of the

revised manuscript under “PJHIC Growth” and highlighted it accordingly:

“The Ni/Co foils were cut into 21 mm × 21 mm pieces... PMMA is chosen as the carbon source suitable for the ultrafast PJHIC process. In the cyclic growth experiments, the PMMA carbon source was coated only once at the initial stage and the PMMA-coated sample was directly placed on graphite paper for the subsequent cyclic graphitization process. ...”

Comment 3:

Although the supplementary information provides a diffusion model based on Fick's laws, a more detailed analysis incorporating the Ni–C phase diagram would strengthen the theoretical foundation. Quantitative comparisons between predicted and experimental segregation thicknesses, along with a discussion of any discrepancies (e.g., due to kinetic limitations or carbon loss), would provide deeper insight into the non-equilibrium segregation mechanism.

Authors' response:

We sincerely appreciate the reviewer's insightful question regarding the quantitative definition of our non-equilibrium process. As rightly pointed out, fast heating/cooling rates are the experimental manifestation, while the underlying physics is rooted in enhanced thermodynamic driving forces. To address this, we have developed a **two-stage model** that quantitatively describes the non-equilibrium segregation process, emphasizing the roles of **interfacial carbon supersaturation (ΔC)** and **phase transformation driving force ($\Delta\mu$)**, as well as compared the **carbon flux** against a traditional isothermal CVD process.

Stage I: Solute depletion-driven carbon segregation to the interface

The PJHIC process initiates with rapid cooling from a starting high temperature (T_0 , 1300°C) to a lower segregation temperature ($1000^\circ\text{C} < T < 1300^\circ\text{C}$). This creates a massive transient supersaturation at the metal-graphite interface. According to the Ni–C phase diagram (Fig. R2-1) and the empirical relationship proposed by Lander &

Marshall [*J. Appl. Phys.* **1952**, 23 (12), 1305]:

$$\ln S = 2.480 - \frac{4880}{T}$$

where S denotes weight-percent carbon solubility (wt %, grams C per 100 grams Ni).

For small values of S , the dimensionless mass fraction C can be directly calculated by:

$$C = \frac{S}{100 + S} \approx \frac{S}{100} = \frac{1}{100} \exp\left(2.480 - \frac{4880}{T}\right)$$

For just 4.0 ~ 4.7 s of cooling time (see comment 4), the equilibrium carbon solubility drops from $C_0(1300^\circ\text{C}) = 0.536$ wt.% to $C_1(1000^\circ\text{C}) = 0.259$ wt.%. The ultrafast cooling rate contributes great interfacial supersaturation $\Delta C = C_0(1300^\circ\text{C}) - C_1(1000^\circ\text{C}) = 0.277$ wt.%, and thus drives a chemical potential difference $\Delta\mu$. The molar Gibbs free energy (μ) is expressed as:

$$\mu(C, T) = \mu^0(T) + k_B T \ln(C)$$

where $\mu^0(T)$ is the standard (reference-state) chemical potential of carbon in Ni at temperature T , k_B is the Boltzmann constant. We assume that the carbon has been solved in Ni bulk at $T_0 = 1300$ °C with an equilibrium carbon solubility $C_0(1300^\circ\text{C}) = 0.536$ wt.%. When the temperature drops, the **excess chemical potential** that drives segregation to the surface at temperature T is:

$$\Delta\mu = \mu(C_0, T) - \mu(C(T), T) = k_B T \ln \frac{C_0}{C(T)}$$

For $T_1 = 1000^\circ\text{C} = 1273$ K, $\Delta\mu \approx 7.73 \times 10^3$ J/mol (≈ 0.080 eV/mol) . The significantly high ΔC and $\Delta\mu$ promote the subsequent phase transformation from carbon atoms to graphite layers.

[Figure Redacted]

Stage II: Supersaturation-driven graphite layer formation

The supersaturated carbon at the interface must overcome the nucleation barrier to form graphite layers. In classical nucleation theory, graphene nucleation on metal surfaces is an activated process in which a stable nucleus forms once it surpasses a critical size, balancing the energetic competition between creating a new interface (surface energy cost) and forming a new phase (volumetric energy gain from supersaturation). A new phase (graphene nucleus) forms via discrete nucleation events rather than continuous transformation. Nuclei are assumed circular (2D disc-shaped) and energetically isotropic. Interface properties (line energy λ) and driving force ($\Delta\mu$) are considered constant and uniform during nucleation. Nucleation is thermodynamically driven and kinetically activated, involving the random fluctuation-driven formation of critical-sized clusters [*Acta Chim. Sin.* **2014**, 72, 345-358; *J. Am. Chem. Soc.* **2011**, 133 (13), 5009-5015].

The Gibbs free energy for forming a circular graphene nucleus of radius r is:

$$\Delta G(r) = 2\pi r\lambda - \pi r^2\rho_g\Delta\mu$$

Where $\lambda \approx 1$ eV/nm is the graphene-Ni line energy [*Nano lett.* **2011**, 11, 518-522], $\rho_g = 3.82 \times 10^{19}$ atom m^{-2} is the areal atomic density, and $\Delta\mu$ is the areal thermodynamic driving force. The critical nucleation radius r^* and barriers ΔG^* follow

from $\partial\Delta G/\partial r = 0$:

$$r^* = \frac{\lambda}{\rho_g \Delta\mu}, \quad \Delta G^* = \frac{\pi\lambda^2}{\rho_g \Delta\mu}$$

We can find that the r^* and ΔG^* both are inversely proportional to $\Delta\mu$. **The high $\Delta\mu$ in our PJHIC process significantly lowers ΔG^* , facilitating high nucleation density and rapid layer formation** (Fig. 1g,h). For nucleation of subsequent layers (2nd layer, 3rd layer,...) beneath the first layer, the same theoretical framework applies, though with a generally lower interfacial line energy λ_2 due to spatial confinement and weaker coupling to metal substrate. Graphene thickening proceeds once the interfacial carbon activity sufficiently reduces ΔG^* small enough.

Comparisons between predicted and experimental thicknesses

Our theoretical analysis is based on the Ni–C phase diagram. At the heating temperature of 1300 °C, the bulk Ni foil (50 μm thick) can reach the equilibrium carbon solubility corresponding to this temperature, which is approximately 0.536 at.% (equivalent to 0.536 wt.%). According to the principle of mass conservation, we assume that all the dissolved carbon atoms segregate completely to both surfaces of the Ni foil during the subsequent forced cooling process to form graphite films. This allows us to theoretically predict the maximum total film thickness achievable in a single PJHIC pulse. The calculation indicates that the theoretical maximum segregation thickness on one side is approximately 1.17 μm .

A quantitative comparison between this theoretical prediction and the experimentally measured saturation thickness obtained from single-pulse experiments (average value: 1.07 μm , see Fig. R1-3) shows that the experimental result approaches the theoretical limit of carbon solubility–driven segregation, being only about 8.5% lower. This close agreement strongly supports the high carbon-utilization efficiency of the PJHIC process.

We further analyzed the ~8.5% deviation and attribute it primarily to carbon loss during polymer pyrolysis. A portion of the PMMA precursor decomposes into gaseous byproducts (e.g., CH_4 , CO_2 , etc.), which are swept away by the high-speed inert gas

flow before sufficient dissolution into the Ni matrix occurs. To explore the practical limit of single-pulse segregation, we precisely controlled the PMMA coating volume using a micropipette. The relevant experimental parameters and measured results are summarized in Table R1.

Fig. R1-3 | Thickness characterization of graphite edges.

Table R1 | Mass change under different PMMA coating amounts.

Number	m_{Ni}/g	V_{PMMA}/mL	$m_{PMMA}/Ni/g$	Δm_{PMMA} (g)	$m_{Gr}/Ni/g$	$\Delta m_{Gr}/g$	The carbon loss rate/%	The average/%	The error/%
1	0.1839	0.2000	0.1941	0.0102	0.1852	0.0013	12.75%		
2	0.1801	0.4000	0.1994	0.0193	0.1822	0.0021	10.88%		
3	0.1855	0.6000	0.2177	0.0322	0.1874	0.0019	5.90%	7.42%	3.73%
4	0.1836	0.8000	0.2232	0.0396	0.1849	0.0013	3.28%		
5	0.1778	1.0000	0.2289	0.0511	0.1800	0.0022	4.31%		

Accordingly, we have revised the manuscript by adding a paragraph at the end of the section “**Pulsed Joule heating activated ultrafast carbon diffusion and segregation**” :

The rapid non-equilibrium segregation growth process comprises two distinct yet interconnected stages (Supplementary Note 3). In Stage I (Solute depletion-driven carbon precipitation), the rapid cooling from 1300 °C to 1000 °C within 4 ~ 5 seconds creates a massive carbon supersaturation ($\Delta C \approx 0.28$ wt.%) due to the drastic reduction in equilibrium solubility dictated by the Ni-C phase diagram³⁴. This corresponds to a chemical potential difference of $\Delta\mu \approx 0.08$ eV/atom. In Stage II, the accumulated

carbon overcomes a drastically reduced nucleation barrier and form graphite layers. Classical nucleation theory indicates that the high $\Delta\mu$ lowers both the critical nucleation radius r^* and energy barrier ΔG^* , facilitating the observed high-density nucleation and rapid graphite layer formation^{48,49}. Quantitatively, the carbon atomic flux in PJHIC reaches $F_{\text{PJHIC}} \approx 2.1 \times 10^{22}$ atoms $\cdot\text{m}^{-2}\cdot\text{s}^{-1}$, surpassing state-of-the-art CVD graphene growth by two orders of magnitude ($F_{\text{CVD}} \approx 3.1 \times 10^{20}$ atoms $\cdot\text{m}^{-2}\cdot\text{s}^{-1}$)^{50,51}. This fundamental difference in mass transport efficiency underscores the paradigm-shifting nature of the non-equilibrium forced segregation mechanism compared to diffusion-limited processes.

The Supplementary Information has also been revised by adding “Supplementary Note 3” and “Supplementary Note 4”:

Supplementary Note 3. two-stage model for non-equilibrium segregation process

A two-stage model that quantitatively describes the non-equilibrium segregation process, emphasizing the roles of interfacial carbon supersaturation (ΔC) and phase transformation driving force ($\Delta\mu$).

Stage I: Solute depletion-driven carbon segregation to the interface

The PJHIC process initiates with rapid cooling from a starting high temperature (T_0 , 1300°C) to a lower segregation temperature ($1000^\circ\text{C} < T < 1300^\circ\text{C}$). This creates a massive transient supersaturation at the metal-graphite interface. According to the Ni–C phase diagram and the empirical relationship (Equation 7) proposed by Lander & Marshall¹, The rapid cooling time within 4.0–4.7 seconds contributes great interfacial supersaturation $\Delta C = C_0(1300^\circ\text{C}) - C_1(1000^\circ\text{C}) = 0.277$ wt.%. Note that, for small values of S , the dimensionless mass fraction C can be directly

calculated by $C = \frac{S}{100+S} \approx \frac{S}{100} = \frac{1}{100} \exp(2.480 - \frac{4880}{T})$.

The ΔC drives a chemical potential difference $\Delta\mu$. The molar Gibbs free energy (μ) is expressed as:

$$\mu(C, T) = \mu^0(T) + k_B T \ln(C) \quad (10)$$

where $\mu^0(T)$ is the standard (reference-state) chemical potential of carbon in Ni at temperature T , k_B is the Boltzmann constant. We assume that the carbon has been solved in Ni bulk at $T_0 = 1300^\circ\text{C}$ with an equilibrium carbon solubility $C_0(1300^\circ\text{C}) = 0.536$ wt.%. When the temperature drops, the

excess chemical potential that drives segregation to the surface at temperature T is:

$$\Delta\mu = \mu(C_0, T) - \mu(C(T), T) = k_B T \ln \frac{C_0}{C(T)} \quad (11)$$

For $T_1 = 1000^\circ\text{C} = 1273\text{ K}$, $\Delta\mu \approx 7.73 \times 10^3\text{ J/mol}$ ($\approx 0.080\text{ eV/mol}$). The significantly high ΔC and $\Delta\mu$ promote the subsequent phase transformation from carbon atoms to graphite layers.

Stage II: Supersaturation-driven graphite layer formation

The supersaturated carbon at the interface must overcome the nucleation barrier to form graphite layers. In classical nucleation theory, graphene nucleation on metal surfaces is an activated process in which a stable nucleus forms once it surpasses a critical size, balancing the energetic competition between creating a new interface (surface energy cost) and forming a new phase (volumetric energy gain from supersaturation). A new phase (graphene nucleus) forms via discrete nucleation events rather than continuous transformation. Nuclei are assumed circular (2D disc-shaped) and energetically isotropic. Interface properties (line energy λ) and driving force ($\Delta\mu$) are considered constant and uniform during nucleation. Nucleation is thermodynamically driven and kinetically activated, involving the random fluctuation-driven formation of critical-sized clusters^{2,3}.

The Gibbs free energy for forming a circular graphene nucleus of radius r is:

$$\Delta G(r) = 2\pi r\lambda - \pi r^2 \rho_g \Delta\mu \quad (12)$$

Where $\lambda \approx 1\text{ eV/nm}$ is the graphene-Ni line energy⁴, $\rho_g = 3.82 \times 10^{19}\text{ atom m}^{-2}$ is the areal atomic density, and $\Delta\mu$ is the areal thermodynamic driving force. The critical nucleation radius r^* and barriers ΔG^* follow from $\partial\Delta G/\partial r = 0$:

$$r^* = \frac{\lambda}{\rho_g \Delta\mu}, \quad \Delta G^* = \frac{\pi\lambda^2}{\rho_g \Delta\mu} \quad (13)$$

We can find that the r^* and ΔG^* both are inversely proportional to $\Delta\mu$. The high $\Delta\mu$ in our PJHIC process significantly lowers ΔG^* , facilitating high nucleation density and rapid layer formation (Fig. 1g,h). For nucleation of subsequent layers (2nd layer, 3rd layer,...) beneath the first layer, the same theoretical framework applies, though with a generally lower interfacial line energy λ_2 due to spatial confinement and weaker coupling to metal substrate. Graphene thickening proceeds once the interfacial carbon activity sufficiently reduces ΔG^* small enough.

Comparison between PJHIC and conventional isothermal CVD

In conventional isothermal CVD at 1000°C , supersaturation is limited and originates not from bulk solubility changes, but from the surface decomposition of hydrocarbon precursors (e.g.,

methane, acetylene). These precursors adsorb and dissociate to form active carbon species, which then migrate to create localized supersaturation. Given this fundamental difference in mechanism, **carbon atomic flux (F)** provides a more direct and quantitative parameter for comparing the mass transport behaviors of CVD and PJHIC processes.

For the PJHIC process, the effective growth duration is within 4s ~ 4.7s, and the vertical growth rate is calculated as 11,162.8 nm min⁻¹ (see Response to Comment 4). The carbon flux is calculated as:

$$F_{PJHIC} = 2.1 \times 10^{22} \text{ atoms}/(m^2 \cdot s) \quad (14)$$

The local concentration of active carbon species in CVD has not been experimentally quantified. So we estimated the effective flux (for lattice incorporation) using data from two benchmark CVD studies: I) the current growth record-holder⁵, II) a high-speed process on CuNi alloy⁶.

$$F_I = 3.06 \times 10^{20} \text{ atoms}/(m^2 \cdot s) \quad (15)$$

$$F_{II} = 1.27 \times 10^{17} \text{ atoms}/(m^2 \cdot s) \quad (16)$$

The calculation results clearly demonstrate that the carbon atomic flux of the PJHIC process ($2.11 \times 10^{22} \text{ atoms m}^{-2} \text{ s}^{-1}$) exceeds that of conventional isothermal CVD processes ($3.06 \times 10^{20} \text{ atoms m}^{-2} \text{ s}^{-1}$) by at least two orders of magnitude, highlighting the exceptional mass transport efficiency and high segregation driving force enabled by the non-equilibrium forced segregation mechanism.

Supplementary Note 4. The single segregation limit of graphite films

By systematically increasing the PMMA coating volume (0.2-0.8 mL) and extending the annealing time to 5 minutes, a maximum experimental thickness of $1.067 \pm 0.032 \mu\text{m}$ was achieved at a coating volume of 0.6 mL. The relationship between film thickness and PMMA dosage exhibited a non-monotonic trend, first increasing and then decreasing. This behavior was attributed to excessive PMMA decomposition, which generated gaseous byproducts (CH₄, C₂H₄, CO, etc.), resulting in substantial carbon loss and thus limiting further thickness growth. Details regarding carbon loss of PMMA please see in Supplementary Table 1.

Comment 4:

The in-plane thermal conductivity of the PJHIC-graphite film (1314 W·m⁻¹·K⁻¹) is notable, but not the highest among reported graphite materials. The authors are encouraged to include a more detailed analysis linking the defect density (e.g., via Raman ID/IG ratio) and grain boundary structure (from EBSD) to the thermal conductivity. A discussion on how further defect reduction or grain orientation optimization could enhance thermal performance would be valuable for readers interested in high-thermal-conductivity applications.

Authors' response:

We sincerely thank the reviewer for this insightful comment. We fully agree that a detailed analysis linking defect density is critical to elucidating the in-plane heat transport mechanisms in our PJHIC- graphite films.

The PJHIC-graphite film exhibits an ID/IG ratio of 0.03 in its Raman spectrum. Using the empirical equation proposed by Pimenta and Dresselhaus et al. for in-plane crystallite size calculation [*Phys. Chem. Chem. Phys.* **9**, 1276-1290 (2007)]:

$$L_a(\text{nm}) = (2.4 \times 10^{-10}) \lambda_{\text{laser}}^4 \left(\frac{I_D}{I_G} \right)^{-1}$$

where λ_{laser} (532 nm) is the laser wavelength, we calculate the in-plane crystallite size L_a of 650 nm. To correlate this with thermal conductivity, we compared our experimental thermal conductivity (1314 W·m⁻¹·K⁻¹) with Fugallo *et al.*'s first-principles calculations², which depict Λ as a function of the effective thermal characteristic length (Fig. R1-4). Fugallo *et al.*'s curve shows a significant decrease in Λ when the characteristic length is below 10 μm . Specifically, their calculation predicts Λ of 1220 W·m⁻¹·K⁻¹ for a characteristic length of 650 nm. The good agreement between our experimental results and first-principles calculations (within expected experimental uncertainties) shows that finite crystallite size is a primary factor limiting further thermal conductivity improvements.

We concur with the reviewer that reducing defects and optimizing grain orientation

have substantial potential to further enhance in-plane thermal conductivity. In future sample fabrication efforts, we will target two key strategies: (1) Reducing defect density and increasing crystallite size by optimizing heating/cooling protocols (e.g., slower heating rates to promote grain growth and rapid quenching to suppress defect formation) and refining carbon dissolution-segregation processes; (2) Improving grain orientation uniformity via enhanced texture control during film formation, which will minimize phonon scattering at grain boundaries. These modifications are expected to push thermal conductivity closer to the theoretical limit, making the material more competitive for high-thermal-conductivity applications.

Fig. R1-4 | In-plane thermal conductivity of graphite as a function of grain size (L_a). TDTR measurement of $\Lambda_{||}$ at 300 K (black squares) and Fugallo *et al.* (orange dashed dot lines) are presented [*Nano Lett.* **14**, 6109-6114 (2014)].

Accordingly, we have revised the manuscript in the conclusion part:

“In summary, ... This approach marks a substantial advance in synthesis speed, but the crystalline perfection of the resulting films still falls short of single-crystal standards, as the rapid non-equilibrium process introduces grain boundaries, point defects, and wrinkles. Our insights into segregation dynamics suggest that further optimization of kinetic control could reconcile both speed and quality.”

Response to the Reviewer #2

This manuscript presents a work on the ultrafast synthesis of high-quality graphite films using a pulsed Joule heating approach. The concept of "non-equilibrium carbon flux engineering" is introduced and investigated through a series of experiments characterizing growth kinetics, structural properties, and thermal performance. The study demonstrates rapid film growth rates and the ability to control film thickness. However, several aspects require further clarification:

Authors' response:

We thank the reviewer for the positive assessment of our work and the insightful comments. We have carefully addressed all points raised, with special attention to clarifying the mechanistic aspects of non-equilibrium carbon transport and segregation. Our point-by-point responses are provided below.

Comment 1:

The core concept revolves around a "non-equilibrium" process, primarily defined by fast heating/cooling rates (>300 °C/s). Beyond the kinetic rates, what are the key physical parameters (e.g., carbon supersaturation at the interface, phase transformation driving force) that quantitatively define this non-equilibrium state? Could the authors provide a quantitative comparison (even a model-based estimation) of the carbon flux or segregation driving force against a traditional isothermal CVD process with a similar effective growth duration (e.g., ~60 s)? This would more forcefully demonstrate the unique advantage of the method beyond simply being "faster."

Authors' response:

We sincerely appreciate the reviewer's insightful question regarding the quantitative definition of our non-equilibrium process. As rightly pointed out, fast

heating/cooling rates are the experimental manifestation, while the underlying physics is rooted in enhanced thermodynamic driving forces. To address this, we have developed a **two-stage model** that quantitatively describes the non-equilibrium segregation process, emphasizing the roles of **interfacial carbon supersaturation (ΔC)** and **phase transformation driving force ($\Delta\mu$)**, as well as compared the **carbon flux** against a traditional isothermal CVD process.

Stage I: Solute depletion-driven carbon segregation to the interface

The PJHIC process initiates with rapid cooling from a starting high temperature (T_0 , 1300°C) to a lower segregation temperature ($1000^\circ\text{C} < T < 1300^\circ\text{C}$). This creates a massive transient supersaturation at the metal-graphite interface. According to the Ni–C phase diagram (Fig. R2-1) and the empirical relationship proposed by Lander & Marshall [*J. Appl. Phys.* **1952**, 23 (12), 1305]:

$$\ln S = 2.480 - \frac{4880}{T}$$

where S denotes weight-percent carbon solubility (wt %, grams C per 100 grams Ni).

For small values of S , the dimensionless mass fraction C can be directly calculated by:

$$C = \frac{S}{100 + S} \approx \frac{S}{100} = \frac{1}{100} \exp\left(2.480 - \frac{4880}{T}\right)$$

For just 4.0 ~ 4.7 s of cooling time (see comment 4), the equilibrium carbon solubility drops from $C_0(1300^\circ\text{C}) = 0.536$ wt.% to $C_1(1000^\circ\text{C}) = 0.259$ wt.%. The ultrafast cooling rate contributes great interfacial supersaturation $\Delta C = C_0(1300^\circ\text{C}) - C_1(1000^\circ\text{C}) = 0.277$ wt.%, and thus drives a chemical potential difference $\Delta\mu$. The molar Gibbs free energy (μ) is expressed as:

$$\mu(C, T) = \mu^0(T) + k_B T \ln(C)$$

where $\mu^0(T)$ is the standard (reference-state) chemical potential of carbon in Ni at temperature T , k_B is the Boltzmann constant. We assume that the carbon has been solved in Ni bulk at $T_0 = 1300^\circ\text{C}$ with an equilibrium carbon solubility $C_0(1300^\circ\text{C}) = 0.536$ wt.%. When the temperature drops, the **excess chemical potential** that drives segregation to the surface at temperature T is:

$$\Delta\mu = \mu(C_0, T) - \mu(C(T), T) = k_B T \ln \frac{C_0}{C(T)}$$

For $T_1 = 1000^\circ\text{C} = 1273$ K, $\Delta\mu \approx 7.73 \times 10^3$ J/mol (≈ 0.080 eV/mol). The

significantly high ΔC and $\Delta\mu$ promote the subsequent phase transformation from carbon atoms to graphite layers.

[Figure Redacted]

Stage II: Supersaturation-driven graphite layer formation

The supersaturated carbon at the interface must overcome the nucleation barrier to form graphite layers. In classical nucleation theory, graphene nucleation on metal surfaces is an activated process in which a stable nucleus forms once it surpasses a critical size, balancing the energetic competition between creating a new interface (surface energy cost) and forming a new phase (volumetric energy gain from supersaturation). A new phase (graphene nucleus) forms via discrete nucleation events rather than continuous transformation. Nuclei are assumed circular (2D disc-shaped) and energetically isotropic. Interface properties (line energy λ) and driving force ($\Delta\mu$) are considered constant and uniform during nucleation. Nucleation is thermodynamically driven and kinetically activated, involving the random fluctuation-driven formation of critical-sized clusters [*Acta Chim. Sin.* **2014**, 72, 345-358; *J. Am. Chem. Soc.* **2011**, 133 (13), 5009-5015].

The Gibbs free energy for forming a circular graphene nucleus of radius r is:

$$\Delta G(r) = 2\pi r\lambda - \pi r^2\rho_g\Delta\mu$$

Where $\lambda \approx 1$ eV/nm is the graphene-Ni line energy [*Nano lett.* **2011**, 11, 518-522],

$\rho_g = 3.82 \times 10^{19} \text{ atom m}^{-2}$ is the areal atomic density, and $\Delta\mu$ is the areal thermodynamic driving force. The critical nucleation radius r^* and barriers ΔG^* follow from $\partial\Delta G/\partial r = 0$:

$$r^* = \frac{\lambda}{\rho_g \Delta\mu}, \quad \Delta G^* = \frac{\pi\lambda^2}{\rho_g \Delta\mu}$$

We can find that the r^* and ΔG^* both are inversely proportional to $\Delta\mu$. **The high $\Delta\mu$ in our PJHIC process significantly lowers ΔG^* , facilitating high nucleation density and rapid layer formation** (Fig. 1g,h). For nucleation of subsequent layers (2nd layer, 3rd layer,...) beneath the first layer, the same theoretical framework applies, though with a generally lower interfacial line energy λ_2 due to spatial confinement and weaker coupling to metal substrate. Graphene thickening proceeds once the interfacial carbon activity sufficiently reduces ΔG^* small enough.

Quantitative comparison between PJHIC and conventional CVD

As the reviewer's suggestion, we compare the driving force and carbon flux between PJHIC and traditional isothermal CVD process with a similar effective growth duration of 5 s.

For the PJHIC process, the effective growth duration is within 4s ~ 4.7s, and the vertical growth rate is calculated as 11,162.8 nm min⁻¹ (see Response to Comment 4). The carbon flux is calculated as:

$$F_{PJHIC} = 2.1 \times 10^{22} \text{ atoms}/(\text{m}^2 \cdot \text{s})$$

In conventional isothermal CVD at 1000°C, supersaturation is limited and originates not from bulk solubility changes, but from the surface decomposition of hydrocarbon precursors (e.g., methane, acetylene). These precursors adsorb and dissociate to form active carbon species, which then migrate to create localized supersaturation. The local concentration of active carbon species in CVD has not been experimentally quantified. So we estimated the effective flux (for lattice incorporation) using data from two benchmark CVD studies: the current growth record-holder (*Nat. Chem.* **2019**, *11* (8), 730-736) and a high-speed process on CuNi alloy (*ACS Nano* **2018**, *12* (6), 6117-6127).

For the literature *Nat. Chem.* **2019**, *11* (8), 730-736, taking the typical data of Fig 1b-d as an example, the substrate area where flux occurs

$$A_{sub} = 1.96 \times 10^{-8} \text{m}^2$$

the growth rate of graphene island (area)

$$\frac{dA_{graphene}}{dt} = \frac{A_{graphene}}{t} = \frac{7.854 \times 10^{-7} \text{m}^2}{5 \text{ s}} = 1.5708 \times 10^{-7} \text{m}^2/\text{s}$$

the carbon atom deposition rate

$$R_{carbon\ atoms} = \rho \frac{dA_{Graphene}}{dt} = 6 \times 10^{12} \text{ atoms/s}$$

Where $\rho = 3.82 \times 10^{19} \text{ atoms/m}^2$ is surface atomic density of graphene. Hence, the carbon flux can be calculated as:

$$F_{Nat.Chem.} = \frac{R_{carbon\ atoms}}{A_{sub}} = 3.06 \times 10^{20} \text{ atoms}/(\text{m}^2 \cdot \text{s})$$

Similarly, for the *ACS Nano* **2018**, *12* (6), 6117-6127,

$$F_{ACS\ Nano} = \rho \times \frac{dA_{graphene}}{dt A_{sub}} \approx 1.27 \times 10^{17} \text{ atoms}/(\text{m}^2 \cdot \text{s})$$

All calculations were conducted using the following standard physicochemical parameters: Graphite density ($\rho = 2.26 \text{ g/cm}^3$); Molar mass of carbon ($M = 12.01 \text{ g/mol}$); Avogadro's number ($N_a = 6.022 \times 10^{23} / \text{mol}$); C–C bond length ($d_{c-c} = 0.142 \text{ nm}$); Single-layer graphene thickness (0.34 nm); Lattice constant ($a = \sqrt{3} \times d_{c-c} \approx 0.246 \text{ nm}$); Unit cell area ($A_{cell} = \frac{\sqrt{3}}{2} \times a^2 \approx 5.246 \times 10^{-20} \text{m}^2$); Surface atomic density ($\rho = \frac{2}{A_{cell}} \approx 3.82 \times 10^{19} \text{ atoms/m}^2$). Here, n_V was denoted as the carbon atomic density.

The calculation results clearly demonstrate that the carbon atomic flux of the PJHIC process ($2.11 \times 10^{22} \text{ atoms m}^{-2} \text{ s}^{-1}$) exceeds that of conventional isothermal CVD processes ($3.06 \times 10^{20} \text{ atoms m}^{-2} \text{ s}^{-1}$) by two orders of magnitude, highlighting the exceptional mass transport efficiency and high segregation driving force enabled by the non-equilibrium forced segregation mechanism.

Accordingly, we have revised the manuscript by adding a paragraph at the end of the section “**Pulsed Joule heating activated ultrafast carbon diffusion and segregation**”:

The rapid non-equilibrium segregation growth process comprises two distinct yet interconnected stages (Supplementary Note 3). In Stage I (Solute depletion-driven carbon precipitation), the rapid cooling from 1300 °C to 1000 °C within 4 ~ 5 seconds creates a massive carbon supersaturation ($\Delta C \approx 0.28$ wt.%) due to the drastic reduction in equilibrium solubility dictated by the Ni-C phase diagram³⁴. This corresponds to a chemical potential difference of $\Delta\mu \approx 0.08$ eV/atom. In Stage II, the accumulated carbon overcomes a drastically reduced nucleation barrier and form graphite layers. Classical nucleation theory indicates that the high $\Delta\mu$ lowers both the critical nucleation radius r^* and energy barrier ΔG^* , facilitating the observed high-density nucleation and rapid graphite layer formation^{48,49}. Quantitatively, the carbon atomic flux in PJHIC reaches $F_{\text{PJHIC}} \approx 2.1 \times 10^{22}$ atoms $\cdot\text{m}^{-2}\cdot\text{s}^{-1}$, surpassing state-of-the-art CVD graphene growth by two orders of magnitude ($F_{\text{CVD}} \approx 3.1 \times 10^{20}$ atoms $\cdot\text{m}^{-2}\cdot\text{s}^{-1}$)^{50,51}. This fundamental difference in mass transport efficiency underscores the paradigm-shifting nature of the non-equilibrium forced segregation mechanism compared to diffusion-limited processes.

The Supplementary Information has also been revised by adding “Supplementary Note 3”:

Supplementary Note 3. two-stage model for non-equilibrium segregation process

A two-stage model that quantitatively describes the non-equilibrium segregation process, emphasizing the roles of interfacial carbon supersaturation (ΔC) and phase transformation driving force ($\Delta\mu$).

Stage I: Solute depletion-driven carbon segregation to the interface

The PJHIC process initiates with rapid cooling from a starting high temperature (T_0 , 1300°C) to a lower segregation temperature ($1000^\circ\text{C} < T < 1300^\circ\text{C}$). This creates a massive transient supersaturation at the metal-graphite interface. According to the Ni-C phase diagram and the empirical relationship (Equation 7) proposed by Lander & Marshall¹, The rapid cooling time within 4.0~4.7 seconds contributes great interfacial supersaturation $\Delta C = C_0(1300^\circ\text{C}) - C_1(1000^\circ\text{C}) = 0.277$ wt.%. Note that, for small values of S , the dimensionless mass fraction C can be directly

$$\text{calculated by } C = \frac{S}{100+S} \approx \frac{S}{100} = \frac{1}{100} \exp\left(2.480 - \frac{4880}{T}\right).$$

The ΔC drives a chemical potential difference $\Delta\mu$. The molar Gibbs free energy (μ) is expressed as:

$$\mu(C, T) = \mu^0(T) + k_B T \ln(C) \quad (10)$$

where $\mu^0(T)$ is the standard (reference-state) chemical potential of carbon in Ni at temperature T , k_B is the Boltzmann constant. We assume that the carbon has been solved in Ni bulk at $T_0 = 1300^\circ\text{C}$ with an equilibrium carbon solubility $C_0(1300^\circ\text{C}) = 0.536$ wt.%. When the temperature drops, the **excess chemical potential** that drives segregation to the surface at temperature T is:

$$\Delta\mu = \mu(C_0, T) - \mu(C(T), T) = k_B T \ln \frac{C_0}{C(T)} \quad (11)$$

For $T_1 = 1000^\circ\text{C} = 1273$ K, $\Delta\mu \approx 7.73 \times 10^3$ J/mol (≈ 0.080 eV/mol). The significantly high ΔC and $\Delta\mu$ promote the subsequent phase transformation from carbon atoms to graphite layers.

Stage II: Supersaturation-driven graphite layer formation

The supersaturated carbon at the interface must overcome the nucleation barrier to form graphite layers. In classical nucleation theory, graphene nucleation on metal surfaces is an activated process in which a stable nucleus forms once it surpasses a critical size, balancing the energetic competition between creating a new interface (surface energy cost) and forming a new phase (volumetric energy gain from supersaturation). A new phase (graphene nucleus) forms via discrete nucleation events rather than continuous transformation. Nuclei are assumed circular (2D disc-shaped) and energetically isotropic. Interface properties (line energy λ) and driving force ($\Delta\mu$) are considered constant and uniform during nucleation. Nucleation is thermodynamically driven and kinetically activated, involving the random fluctuation-driven formation of critical-sized clusters^{2,3}.

The Gibbs free energy for forming a circular graphene nucleus of radius r is:

$$\Delta G(r) = 2\pi r \lambda - \pi r^2 \rho_g \Delta\mu \quad (12)$$

Where $\lambda \approx 1$ eV/nm is the graphene-Ni line energy⁴, $\rho_g = 3.82 \times 10^{19}$ atom m^{-2} is the areal atomic density, and $\Delta\mu$ is the areal thermodynamic driving force. The critical nucleation radius r^* and barriers ΔG^* follow from $\partial\Delta G/\partial r = 0$:

$$r^* = \frac{\lambda}{\rho_g \Delta\mu}, \quad \Delta G^* = \frac{\pi \lambda^2}{\rho_g \Delta\mu} \quad (13)$$

We can find that the r^* and ΔG^* both are inversely proportional to $\Delta\mu$. The high $\Delta\mu$ in our PJHIC process significantly lowers ΔG^* , facilitating high nucleation density and rapid layer formation (Fig. 1g,h). For nucleation of subsequent layers (2nd layer, 3rd layer,...) beneath the first

layer, the same theoretical framework applies, though with a generally lower interfacial line energy λ_2 due to spatial confinement and weaker coupling to metal substrate. Graphene thickening proceeds once the interfacial carbon activity sufficiently reduces ΔG^* small enough.

Comparison between PJHIC and conventional isothermal CVD

In conventional isothermal CVD at 1000 °C, supersaturation is limited and originates not from bulk solubility changes, but from the surface decomposition of hydrocarbon precursors (e.g., methane, acetylene). These precursors adsorb and dissociate to form active carbon species, which then migrate to create localized supersaturation. Given this fundamental difference in mechanism, ***carbon atomic flux (F)*** provides a more direct and quantitative parameter for comparing the mass transport behaviors of CVD and PJHIC processes.

For the PJHIC process, the effective growth duration is within 4s ~ 4.7s, and the vertical growth rate is calculated as 11,162.8 nm min⁻¹ (see Response to Comment 4). The carbon flux is calculated as:

$$F_{PJHIC} = 2.1 \times 10^{22} \text{ atoms}/(m^2 \cdot s) \quad (14)$$

The local concentration of active carbon species in CVD has not been experimentally quantified. So we estimated the effective flux (for lattice incorporation) using data from two benchmark CVD studies: I) the current growth record-holder⁵, II) a high-speed process on CuNi alloy⁶.

$$F_I = 3.06 \times 10^{20} \text{ atoms}/(m^2 \cdot s) \quad (15)$$

$$F_{II} = 1.27 \times 10^{17} \text{ atoms}/(m^2 \cdot s) \quad (16)$$

The calculation results clearly demonstrate that the carbon atomic flux of the PJHIC process ($2.11 \times 10^{22} \text{ atoms m}^{-2} \text{ s}^{-1}$) exceeds that of conventional isothermal CVD processes ($3.06 \times 10^{20} \text{ atoms m}^{-2} \text{ s}^{-1}$) by at least two orders of magnitude, highlighting the exceptional mass transport efficiency and high segregation driving force enabled by the non-equilibrium forced segregation mechanism.

Comment 2:

The choice of an extremely fast cooling rate (>50 °C/s) and the specific durations for the heating (120 s) and cooling (30 s) intervals in the cyclic process are critical to the

method's success. However, the rationale behind selecting these specific parameters is not clearly explained. Were these values optimized empirically, or are they based on a theoretical consideration (e.g., a characteristic time for carbon diffusion vs. segregation)? Please clarify the reasoning behind these key process choices.

Authors' response:

We sincerely appreciate the reviewer's insightful question concerning the key process parameters. The selection of the specific conditions 120 s for heating and 30 s for cooling was determined through a combination of theoretical considerations, equipment capability, and experimental validation.

1. Heating stage (120 s): Ensuring bulk-phase carbon saturation

The duration of the heating stage was determined by combining the theoretical diffusion requirement of carbon in Ni with experimental observations.

According to our Supplementary Note 1, at the growth temperature $T = 1573$ K, the diffusion coefficient D was calculated to be $6.571 \times 10^{-6} \text{ cm}^2 \cdot \text{s}^{-1}$, and the diffusion time t through a $50 \text{ }\mu\text{m}$ -thick nickel foil was determined to be approximately 3.8 s. Our ToF-SIMS experiments also revealed that the diffusion time is about 11 seconds. However, for the cyclic process, we **should ensure sufficient dissolution of carbon atoms**, thereby achieving a maximized supersaturation state and subsequent forced segregation. According to our experiment in Figure 1 g (Fig. R2-5), the thickness of the graphite film gets saturated after 80 s (by applying $1300 \text{ }^\circ\text{C}$).

Fig. R2-5 (Figure 1g) | Graphite thickness as a function of growth time.

In another hand, for the cyclic process, the growth temperature was set to 1250-1280 °C rather than 1300 °C in order to avoid heat accumulation during repeated cycling. Experimentally, we confirmed this phenomenon that the peak temperature of the system would gradually increase with the number of cycles, potentially causing melting or structural degradation of the Ni foil (Fig. R2-3). Therefore, the applied current was appropriately reduced to balance the Joule heating effect with the system's heat accumulation behavior.

Fig. R2-3 | Optical image of the Ni foil obtained after the 1300 °C cyclic growth process.

Therefore, during the heating stage, a current of 320-325 A was applied to the graphite plate and maintained for 120 s, enabling the system to rapidly reach 1250-1280 °C.

Fig. R2-2 (Supplementary Fig. 9) | The curve of current and temperature varying with time during the cyclic heating and cooling process.

2. Cooling stage (30 s): Balancing forced segregation and system capability

The selection of the cooling time represents a balance between process requirements and system limitations. Process requirement: The cooling stage serves as

the critical window during which “forced segregation” occurs. The system must be rapidly cooled to a temperature where the solubility of carbon in nickel becomes extremely low (well below 1000 °C), in order to induce a strong segregation driving force.

System capability: In our water-cooled system, a 30 s duration corresponds to the typical time required for the temperature to naturally decrease from 1250-1280 °C to below 800 °C after power shutdown. This cooling window has been experimentally verified to be sufficient for completing the main segregation process within a single cycle.

Accordingly, we have revised the manuscript by adding sentences in Methods section as follow:

“The Ni/Co foils were cut into 21 mm × 21 mm pieces... For the cyclic process, a current of 320-325 A was applied to the graphite plate for 120 s to rapidly achieve 1250-1280 °C, ensuring sufficient dissolution and minimizing heat accumulation during repeated cycling. The cooling stage was set to be 30 s to promote effective carbon segregation.”

Comment 3:

The comparison in Figure 1h is excellent, but it primarily includes works prior to 2023. Please clearly articulate the fundamental distinctions between your work and very recent (or contemporary) studies also focusing on rapid synthesis of high-quality graphite films, such as Zhang et al. Nat. Commun. 16, 7180 (2025) (Ref. 21). Given that Ref. 21 also reports considerable growth rate (6.2 layers/s) and high thermal conductivity, what is the definitive advantage of your method in terms of growth speed, energy consumption, or process simplicity?

Authors' response:

We appreciate the reviewer's suggestion to include a comparison with this recent

and important work. We fully acknowledge the significant progress achieved by Zhang *et al.* (Ref. 21) in the fabrication of high-quality graphite films. It should be noted, however, that the PJHIC approach developed in this study differs fundamentally from their “isothermal carbon diffusion/lamination-graphitization” pathway in terms of core mechanism, process design, and product morphology. The PJHIC method demonstrates clear advantages in the preparation efficiency, energy consumption, and procedural simplicity for micrometer-thick films.

To enable a fair and quantitative comparison, the layer growth rate reported in Ref. 21 (6.2 layers s⁻¹) was converted to the vertical growth rate used in this work. Assuming a single graphene layer thickness of 0.335 nm, the equivalent vertical growth rate is approximately 126 nm min⁻¹. In contrast, the PJHIC process developed here achieves a vertical growth rate of 730 nm min⁻¹, which is about 5.8 times higher.

Fig. R2-6 (Revised Fig. 1 h) | Comparative analysis of the growth time and growth rate of graphite films reported in this work and in literatures.

Efficiency and speed advantages for thick-film fabrication: For the preparation of micrometer-scale graphite films (e.g., 5 μm), the layer-by-layer growth and subsequent processing reported in Ref. 21 require a relatively long total time. In contrast, the PJHIC method enables rapid thickening at the bulk-material level, allowing a 5 μm-thick film to be obtained directly within approximately 2 h, thus demonstrating a clear advantage in overall time efficiency.

Low energy consumption and procedural simplicity: The PJHIC approach avoids the high energy consumption associated with conventional CVD processes and long-

duration high-temperature graphitization furnaces. Moreover, it integrates the traditionally multi-step processes into a single-step synthesis, significantly reducing both equipment requirements and procedural complexity.

Therefore, this work not only provides a faster route for graphite film synthesis, but more importantly, it establishes an innovative pathway for high-throughput, low-energy, single-step preparation of micrometer-thick, high-quality graphite films, complementing the precise layer-by-layer epitaxial growth techniques.

In the revised manuscript, Figure 1 has been updated.

Fig. 1 | Ultrafast growth of graphite films via the PJHIC process.

a-c Schematic of the PJHIC system (a), the thermal shock generated by pulsed current application (b), and carbon diffusion-graphite segregation process (c). **d,e** Optical images of the as-synthesized graphite film transferred on Si/SiO₂ (d) and flexible PET (e) substrate, respectively. **f** Current-time and temperature-time profiles corresponding

to different growth durations. **g** Graphite thickness as a function of growth time. Inset: AFM image at side of a typical graphite sheet grown with 72s. **h** Comparative analysis of the growth time and growth rate of graphite films reported in this work and in literatures^{15,20-29,35-42}. **i** Raman spectra acquired at as-synthesized graphite films with different growth time. **j** Deconvolution of a typical 2D Raman band into two Lorentzian components. **k** The statistical distribution of I_D/I_G ratio and FWHM of the G band over an 80 × 80 μm² region.

Comment 4:

The "effective growth time" is defined as the duration above 1000°C, leading to a vertical growth rate of 730 nm/min. However, the manuscript correctly identifies the cooling phase as the primary period for forced segregation. Does this "effective growth time" accurately represent the actual time window for segregation? If segregation occurs predominantly during the few seconds of cooling from 1300°C to 1000°C, the instantaneous segregation rate would be extraordinarily high. This point requires clarification.

Authors' response:

We sincerely thank the reviewer for this insightful comment and important suggestion. The primary driving force for the forced segregation process is indeed concentrated within the short cooling window from 1300 °C to 1000 °C. Accordingly, we have re-evaluated the instantaneous segregation rate specifically during this critical period.

The recalculated cooling durations for the series of experiments (corresponding to Figure 1f, g and Supplementary Fig. 2-3) are presented in Fig. R2-7. The instantaneous segregation rates during this cooling phase are calculated as follows:

For the sample under 11 s growth time, the cooling time $t_1 = 4.7$ s, the instantaneous segregation rate

$$v_1 = \frac{62 \text{ nm}}{4.7 \text{ s}} = 13.2 \text{ nm/s} = 792 \text{ nm/min}$$

For the sample under 42 s growth time, the cooling time $t_2 = 4.6$ s, the

instantaneous segregation rate

$$v_2 = \frac{487 \text{ nm}}{4.6 \text{ s}} = 105.9 \text{ nm/s} = 6,354 \text{ nm/min}$$

For the sample under 51 s growth time, the cooling time $t_3 = 4.3 \text{ s}$, the instantaneous segregation rate

$$v_3 = \frac{608 \text{ nm}}{4.3 \text{ s}} = 141.4 \text{ nm/s} = 8,484 \text{ nm/min}$$

For the sample under 61 s growth time, the cooling time $t_4 = 4.5 \text{ s}$, the instantaneous segregation rate

$$v_4 = \frac{743 \text{ nm}}{4.5 \text{ s}} = 165.1 \text{ nm/s} = 9,906 \text{ nm/min}$$

For the sample under 72 s growth time, the cooling time $t_5 = 4.0 \text{ s}$, the instantaneous segregation rate

$$v_5 = \frac{811 \text{ nm}}{4.0 \text{ s}} = 202.8 \text{ nm/s} = 12,168 \text{ nm/min}$$

For the sample under 102 s growth time, the cooling time $t_6 = 4.2 \text{ s}$, the instantaneous segregation rate

$$v_6 = \frac{834 \text{ nm}}{4.2 \text{ s}} = 198.6 \text{ nm/s} = 11,916 \text{ nm/min}$$

For the sample under 132 s growth time, the cooling time $t_7 = 4.1 \text{ s}$, the instantaneous segregation rate

$$v_7 = \frac{842 \text{ nm}}{4.1 \text{ s}} = 205.4 \text{ nm/s} = 12,324 \text{ nm/min}$$

Fig. R2-7 (Revised Supplementary Fig. 4) | Analysis of cooling duration and corresponding instantaneous segregation rates. The specific cooling time windows

(from 1300 °C to 1000 °C) associated with the temperature profiles in Fig. 1f are delineated.

As these calculations demonstrate, the instantaneous segregation rate during the 4-5 seconds of cooling is extraordinarily high, reaching up to ~12,000 nm/min. However, the complete graphite formation process in the PJHIC strategy comprises two essential stages: (1) the heating/dissolution stage for carbon source decomposition and carbon influx into the metal bulk, and (2) the cooling/segregation stage for graphite formation. The "effective growth time" (duration above 1000 °C) encompasses both stages and represents the total processing time from the system's perspective. Therefore, the **average vertical growth rate of 730 nm/min** (calculated based on the total effective time) remains a meaningful and conservative metric for comparing the overall synthesis efficiency against other methods in the literature.

To provide a comprehensive picture, we have now clarified this distinction in the revised manuscript. We report both the average growth rate (for cross-method comparison) and the instantaneous segregation rate (to highlight the exceptional kinetics of the non-equilibrium forced segregation), as reflected in the added sentences in the main text:

“To investigate the growth rate limit of graphite segregation through Ni foil..... Notably, considering that graphite segregation occurs predominantly during the rapid cooling stage (from 1300 °C to 1000 °C within 4 ~ 5 seconds), the corresponding instantaneous segregation rate is calculated to be ~12,000 nm/min (Supplementary Fig. 4). The PJHIC strategy enables micron-thick, uniform graphite films with growth rates over an order of magnitude higher than previous reports^{15,20-29,35-42}.”

Accordingly, we revised the Supplementary Information by adding Supplementary Fig. 4:

Supplementary Fig. 1 | Analysis of cooling duration and corresponding instantaneous segregation rates. The specific cooling time windows (from 1300 °C to 1000 °C) associated with the temperature profiles in Fig. 1f are delineated. the instantaneous segregation rates are calculated to be: $v_1 = 792$ nm/min, $v_2 = 6,354$ nm/min, $v_3 = 8,484$ nm/min, $v_4 = 9,906$ nm/min, $v_5 = 12,168$ nm/min, $v_6 = 11,916$ nm/min, and $v_7 = 12,324$ nm/min.

Comment 5:

The use of PMMA as a solid-state carbon source is a key experimental detail. What is the specific advantage of using a polymer precursor like PMMA over others carbon sources (e.g., Graphite plate) in this particular PJHIC process? Does its decomposition kinetics at ultra-high heating rates favor a rapid and massive carbon influx into the Ni substrate, potentially creating a higher initial supersaturation? A brief justification for this choice would be helpful.

Authors' response:

We sincerely appreciate the reviewer's insightful question regarding the choice of carbon source. The selection of PMMA was made after systematic experimental comparison and careful mechanistic consideration. The key advantage of PMMA lies

in its decomposition kinetics, which are well matched to the ultrafast heating profile employed in the PJHIC process. This compatibility enables the rapid release of a sufficient amount of carbon atoms within an extremely short time window, thereby providing the necessary conditions for achieving high interfacial carbon supersaturation.

To comprehensively evaluate the influence of different carbon sources, we conducted a series of comparative experiments involving five solid carbon precursors: graphite paper, carbon black powder, polyimide (PI) film, and PMMA. The macroscopic results (Fig. R2-8) clearly demonstrate that among all tested precursors, only PMMA produced a complete, continuous, and uniform graphite film on the Ni foil surface. In contrast, the other carbon sources yielded only discontinuous and patchy graphite domains, failing to achieve full substrate coverage.

Fig. R2-8 | Optical images of samples obtained from different carbon sources. a, graphite paper. **b,** carbon black powder. **c,** PI film. **d,** PMMA.

PMMA exhibits a relatively low onset temperature for pyrolysis (approximately 400 °C) and undergoes rapid, continuous decomposition during heating up to the peak temperature (1300 °C), generating a large quantity of reactive carbon species. Such a precursor-type decomposition behavior is highly compatible with the ultrafast heating rate adopted in PJHIC. Consequently, carbon atoms can be released and dissolved rapidly around the catalytic graphitization temperature of Ni foil. As suggested by the reviewer, this process facilitates the rapid establishment of a high carbon supersaturation within the metal substrate, thereby providing sufficient driving force for the subsequent non-equilibrium enforced segregation.

Graphite paper and carbon black are both thermodynamically stable allotropes of carbon, where carbon atoms can only enter the metallic bulk through a sublimation

mechanism. This process proceeds extremely slowly at 1300 °C and cannot supply sufficient carbon flux within the second-to-minute timescale of PJHIC. As a result, only sparse and discontinuous graphite islands are formed.

Although PI film is also a polymeric precursor, it possesses higher thermal stability and graphitization temperature. Its decomposition kinetics are slower compared to PMMA, leading to an insufficient rate and amount of carbon release under ultrafast heating conditions. This limitation prevents homogeneous nucleation and continuous growth across the entire substrate.

Therefore, the selection of PMMA is not arbitrary, but rather dictated by its ability to meet the key requirement of PJHIC for an extremely high initial carbon flux. Its rapid decomposition kinetics ensure the instantaneous and abundant release of carbon, which serves as the prerequisite for establishing a high initial supersaturation and achieving ultrafast, uniform graphite growth under non-equilibrium conditions.

Accordingly, we have revised the manuscript at the Section of “Cyclic saturation engineering for thickness control” and Methods section:

“Cyclic saturation engineering for thickness control

To transcend the sub-micrometer.....Although the 7.42% carbon loss happens due to PMMA decomposition into gaseous byproducts (Supplementary Note 3 and Supplementary Table 1), the decomposed gaseous carbon source favors a rapid and massive carbon influx into the Ni substrate, potentially creating a higher initial supersaturation....30 s of forced cooling for triggering the segregation (Supplementary Fig. 9).”

Revised Methods section, PJHIC growth:

“The Ni/Co foils were cut into 21 mm × 21 mm pieces. Subsequently, approximately 0.6 mL of PMMA solution was spin-coated onto the surface of the foils. Based on a systematic evaluation of decomposition kinetics and comparative experiments with alternative sources (*e.g.*, graphite paper, carbon black powder, and PI film), PMMA is chosen as the carbon source suitable for the ultrafast PJHIC process. ...”

Comment 6:

The rapid thermal shocks and the associated large thermal stress could potentially introduce more wrinkles and defects into the graphite films. The conclusion states that "the crystalline perfection of the resulting films still falls short of single-crystal standards." While this responsible statement is appreciated, it seems somewhat contradictory to the data presented (millimeter-scale grains, excellent Raman signals, ABA stacking). Please specify in what aspects it "falls short." Is it a larger mosaic spread? Or the presence of some stacking faults?

The color bar of Raman map in Supplementary Fig. 4 represents the ID/IG ratio, ranging from 0 to 2, which convincingly shows the overall high quality. However, to honestly reveal and discuss any potential micro-scale heterogeneity or defect populations introduced by the process, would it be more informative to use a narrower color scale for this mapping? The current scale might be masking subtle variations.

Authors' response:

We sincerely thank the reviewer for the insightful comments and constructive suggestions. The reviewer has astutely identified the nuanced structural characteristics of our PJHIC-synthesized graphite films. We appreciate the opportunity to clarify in what specific aspects the crystalline perfection falls short of single-crystal standards.

Indeed, while our films exhibit excellent macroscopic crystallinity (millimeter-scale grains, high-quality Raman signals, ABA stacking), the ultrafast non-equilibrium growth process inherently introduces specific microstructural imperfections that prevent them from achieving true single-crystal perfection. The primary limitations are as follows:

1) **Point defects from rapid kinetics:** The extremely rapid segregation and cooling process intrinsically limits the time for carbon atoms to anneal into the most stable lattice positions. This results in a low, yet non-negligible, density of point defects, as consistently reflected in our Raman statistical analysis (average $I_D/I_G \approx 0.03$, Figure 1k, Figure R2-9 and Revised Supplementary Fig 5).

2) **Wrinkles induced by thermal stress:** The large thermal shocks and associated mismatch stress during rapid cooling cannot be fully relieved, leading to the formation of wrinkles. These are visible in optical micrographs (Figure 1d), SEM images (Supplementary Fig 11f), and the revised Raman mapping data (Revised Supplementary Fig.5). These are a direct consequence of the non-equilibrium process conditions.

3) **Polycrystallinity and grain boundaries:** The graphite films are synthesized on polycrystalline Ni foils. Consequently, the segregated films are inherently polycrystalline as well, albeit with very large grain sizes (millimeter-scale) (Figure 4a,b, Supplementary Fig. 11 and Fig. R2-10). The presence of these grain boundaries inevitably introduces additional scattering centers for phonons, which is a key factor limiting the in-plane thermal conductivity of our films ($1314 \text{ W m}^{-1} \text{ K}^{-1}$) from reaching the values of the highest-grade single-crystal HOPG ($>2000 \text{ W m}^{-1} \text{ K}^{-1}$).

4) **Presence of stacking faults:** Regarding stacking faults, while a detailed statistical analysis of their density is beyond the scope of this current work and will be a focus of future studies, the high quality of our XRD data (Figure 4c) showing an interlayer spacing (3.355 \AA) and a sharp (0002) peak ($\text{FWHM} = 0.169^\circ$) comparable to those of high-quality HOPG and Kish graphite—indicates that the overall stacking order is predominantly the thermodynamically stable ABA type.

Following the reviewer's valuable suggestion, we have revised the Raman mapping in Supplementary Fig. 4 by narrowing the I_D/I_G color scale from 0–2 to 0–0.5. This adjustment, as shown in the updated figure below, effectively reveals the previously masked micro-scale heterogeneity. It now clearly demonstrates that: 1) The basal plane of the large grains exhibits very low defect levels ($I_D/I_G \approx 0.03$), corresponding to the point defect density mentioned above; 2) The defect ratio increases significantly (> 0.1 , locally > 0.2) precisely along the wrinkle lines, directly validating the spatial correlation between these structural features and enhanced defect formation.

Fig. R2-9 (Revised Supplementary Fig. 5) | Raman characterization of graphite films. (a) Typical Raman spectra of the graphite film at 6 random positions. (b-d) Raman mapping of the graphite film showing I_D/I_G (b), FWHM of the G band (c), and peak position of $2D_2$ (d).

Fig. R2-10 | Grain size of graphite films. (a, b) EBSD inverse pole figure mappings along the Z-direction and Y-direction of graphite films grown by segregation on the surface of ordinary commercial nickel foils.

Accordingly, we have revised the manuscript in the conclusion part by specify the

“fall short” aspects:

“In summary, ... This approach marks a substantial advance in synthesis speed, but the crystalline perfection of the resulting films still falls short of single-crystal standards, as the rapid non-equilibrium process introduces grain boundaries, point defects, and wrinkles. Our insights into segregation dynamics suggest that further optimization of kinetic control could reconcile both speed and quality...”

In all, we appreciate having highly constructive remarks from all reviewers. With all their comments addressed in the text, we are looking forward to the publication of our manuscript in *Nature communication*.

Sincerely yours,

Zhongfan Liu and Luzhao Sun

List of changes

1. Enhanced theoretical foundation: A comprehensive two-stage model for the non-equilibrium segregation process has been developed and added to the manuscript (end of section "Pulsed Joule heating activated ultrafast carbon diffusion and segregation") and the Supplementary Information (as Supplementary Note 3).
2. Clarification of process parameters: The rationale behind key experimental parameters, including the selection of heating (120 s) and cooling (30 s) durations in the cyclic process, has been clarified in the Methods section. A justification for choosing PMMA as the carbon source, based on its decomposition kinetics and comparative experiments with other sources (graphite paper, carbon black, PI film), has been added to the "Cyclic saturation engineering" section and the Methods section.
3. Demonstration of scalability: Data demonstrating the synthesis of a larger-area graphite film (12 cm × 5 cm) has been added as Revised Supplementary Fig. 16.
4. Refined growth rate analysis: The concept and calculation of the instantaneous segregation rate (reaching up to ~12,000 nm/min during the rapid cooling phase) have been introduced. This analysis, along with the corresponding cooling time windows, is now presented in the main text and as Revised Supplementary Fig. 4.
5. In-depth analysis of material quality: A more detailed discussion linking defect density (via Raman I_D/I_G ratio and crystallite size calculation) and grain boundary structure to the measured thermal conductivity has been added. The Raman I_D/I_G mapping color scale has been narrowed (0–0.5) in Revised Supplementary Fig. 5 to better reveal micro-scale heterogeneity.
6. The Figure number in main text and supplementary information have been updated.
7. Considering the contribution regarding the theoretical discussions on the segregation model, we have added two authors, Xiongzhi Zeng and Zhenyu Li.

This manuscript presents a work on the ultrafast synthesis of high-quality graphite films using a pulsed Joule heating approach. The concept of "non-equilibrium carbon flux engineering" is introduced and investigated through a series of experiments characterizing growth kinetics, structural properties, and thermal performance. The study demonstrates rapid film growth rates and the ability to control film thickness. However, several aspects require further clarification:

Q1: The core concept revolves around a "non-equilibrium" process, primarily defined by fast heating/cooling rates (>300 °C/s). Beyond the kinetic rates, what are the key physical parameters (e.g., carbon supersaturation at the interface, phase transformation driving force) that quantitatively define this non-equilibrium state? Could the authors provide a quantitative comparison (even a model-based estimation) of the carbon flux or segregation driving force against a traditional isothermal CVD process with a similar effective growth duration (e.g., ~ 60 s)? This would more forcefully demonstrate the unique advantage of the method beyond simply being "faster."

Q2: The choice of an extremely fast cooling rate (>50 °C/s) and the specific durations for the heating (120 s) and cooling (30 s) intervals in the cyclic process are critical to the method's success. However, the rationale behind selecting these specific parameters is not clearly explained. Were these values optimized empirically, or are they based on a theoretical consideration (e.g., a characteristic time for carbon diffusion vs. segregation)? Please clarify the reasoning behind these key process choices.

Q3: The comparison in Figure 1h is excellent, but it primarily includes works prior to 2023. Please clearly articulate the fundamental distinctions between your work and very recent (or contemporary) studies also focusing on rapid synthesis of high-quality graphite films, such as Zhang et al. Nat. Commun. 16, 7180 (2025) (Ref. 21). Given that Ref. 21 also reports considerable growth rate (6.2 layers/s) and high thermal conductivity, what is the definitive advantage of your method in terms of growth speed, energy consumption, or process simplicity?

Q4: The "effective growth time" is defined as the duration above 1000°C , leading to a vertical growth rate of 730 nm/min. However, the manuscript correctly identifies the cooling phase as the primary period for forced segregation. Does this "effective growth time" accurately represent the actual time window for segregation? If segregation occurs predominantly during the few seconds of cooling from 1300°C to 1000°C , the instantaneous segregation rate would be extraordinarily high. This point requires clarification.

Q5: The use of PMMA as a solid-state carbon source is a key experimental detail. What is the specific advantage of using a polymer precursor like PMMA over others carbon sources (e.g., Graphite plate) in this particular PJHIC process? Does its decomposition kinetics at ultra-high heating rates favor a rapid and massive carbon influx into the Ni substrate, potentially creating a higher initial supersaturation? A brief justification for this choice would be helpful.

Q6: The rapid thermal shocks and the associated large thermal stress could potentially introduce more wrinkles and defects into the graphite films. The conclusion states that "the crystalline perfection of the resulting films still falls short of single-crystal standards." While this responsible statement is appreciated, it seems somewhat contradictory to the data presented (millimeter-scale grains, excellent Raman signals, ABA stacking). Please specify in what aspects it "falls short." Is it a larger mosaic spread? Or the presence of some stacking faults? The color bar of Raman map in Supplementary Fig. 4 represents the ID/IG ratio, ranging from 0 to 2, which convincingly shows the overall high quality. However, to honestly reveal and discuss any potential micro-scale heterogeneity or defect populations introduced by the process, would it be more informative to use a narrower color scale for this mapping? The current scale might be masking subtle variations.